# Historical modelling of changes in Lake Erken thermal conditions

Simone Moras, Ana I. Ayala, Don C. Pierson

Department of Ecology and Genetics, Uppsala University, Uppsala, 75236, Sweden

*Correspondence to*: Simone Moras (simone.moras@ebc.uu.se)

**Abstract.** Historical lake water temperature records are a valuable source of information to assess the influence of climate change on lake thermal structure. However, in most cases such records span a short period of time and/or are incomplete, providing a less credible assessment of change. In this study, the hydrodynamic model GOTM (General Ocean Turbulence Model, a hydrodynamic model configured in Lake Mode) was used to reconstruct daily profiles of water temperature in Lake Erken (Sweden) over the period 1961-2017, using seven climatic parameters as forcing data: wind speed (WS), air temperature

(Air T), atmospheric pressure (Air P), relative humidity (RH), cloud cover (CC), precipitation (DP) and shortwave radiation (SWR). The model was calibrated against observed water temperature data collected during the study interval, and the calibrated model revealed a good match between modelled and observed temperature (RMSE=1.089 °C). From the long-term simulations of water temperature, this study focused on detecting possible trends in water temperature over the entire study interval 1961-2017 and in the sub-intervals 1961-1988 and 1989-2017, since an abrupt change in air temperature was detected

in 1988. The analysis of the simulated temperature showed that epilimnetic temperature has increased on average by 0.444°C/decade and 0.792 °C/decade in spring and autumn in the sub-interval 1989-2017. Summer epilimnetic temperature has increased by 0.351 °C/decade over the entire interval 1961-2017. Hypolimnetic temperature has increased significantly in spring over the entire interval 1961-2017 by 0.148 °C/decade and by 0.816 °C/decade in autumn in the sub-interval 1989-2016. Whole-lake temperature showed a significant increasing trend in the sub-interval 1989-2017 during spring (0.404

°C/decade), and autumn (0.789 °C/decade, interval 1989-2016), while a significant trend was detected in summer over the entire study interval 1961-2017 (0.239 °C/decade). Moreover, this study showed that changes in the phenology of thermal stratification have occurred over the 57-years period of study. Since 1961, the stability of stratification (Schmidt Stability) has increased by 5.365 Jm$^{-2}$/decade. The duration of thermal stratification has increased by 7.297 days/decade, correspondent with an earlier onset of stratification of ~ 16 days and to a delay of stratification termination of ~ 26 days. The average thermocline

depth during stratification became shallower by ~1.345 m, and surface-bottom temperature difference increased over time by 0.249 °C/decade. The creation of daily-time step water temperature dataset not only provided evidence of changes in Erken thermal structure over the last decades, but it is also a valuable resource of information that can help in future research on the ecology of Lake Erken. The use of readily available meteorological data to reconstruct Lake Erken's past water temperature is shown to be a useful method to evaluate long-term changes in lake thermal structure, and it is a method that can be extended

to other lakes.

# 1. Introduction

Changes in the thermal structure and mixing regimes of lakes are connected to changes in several climatic factors such as air temperature, solar radiation, cloud cover, wind speed and humidity (Woolway and Merchant, 2019). The alteration of lake hydrodynamic properties has consequences on lake chemistry, biology and ultimately on the ecosystem services that lakes provide (Adrian et al., 2009; Vincent, 2009). Since climatic conditions have changed markedly in the last century and they are expected to change considerably in the next decades (IPCC, 2013), the importance of evaluating how freshwater bodies are affected by climate change becomes evident. A direct assessment of how lakes have already been affected by climate change is to analyse historical trends in lake water temperature data. However, the availability of long-term data of lake water temperature is still scarce. For example, there are very few lakes around the world with a long-term record (defined here as >50 years) of water temperature profiles (e.g. Jankowski et al., 2006; Skowron, 2017). Instead, the availability of long-term historical data is often limited to surface water temperature of one or few lakes (e.g. Livingstone and Dokulil, 2001; Kainz et al., 2017) and the time frame of surface temperature data available for the majority of lakes is limited to 2-3 decades. For example, Sharma et al. (2015) compiled a worldwide database with lake surface water temperature between 1985-2009. The same time frame was used by Schneider and Hook (2010) that reported an average warming trend of $0.045 \pm 0.011°C/year$ of lake surface water temperature in 167 large lakes (>500 km$^2$) using satellite-derived measurements; similarly O'Reilly et al. (2015) reported an average warming trend of 0.34 °C/decade for lake summer surface water temperature in 235 lake worldwide retrieved from both in-situ and satellite data. Even though these studies have demonstrated a rapid warming trend among lakes, the analysis of only surface water temperature is not sufficient to obtain a complete evaluation of the changes in the thermal structure that encompass, for example, temperature trends in the water column and phenology of thermal stratification. Moreover, the scarcity of water temperature data before 1980s makes difficult to assess earlier thermal conditions for the majority of lakes. A longer record of historical data provides more background information, allows better documentation of the changes that have already taken place, and leads to more accurate predictions of lake thermal conditions in future decades. One of the best arguments to counter climate change sceptics is well documented long-term records of the ongoing effects of climate change.

For this reason, the aim of this study was to use a hydrodynamic model to extend records of lake water temperature back in time, in order to provide a longer and more consistent picture of the changes in thermal structure of lakes. We tested this approach on Lake Erken, which has been studied extensively over the past 70 years (Pettersson, 2012). Automated hourly measurements of water temperature and meteorological data have been collected from the lake since October 1988. Before 1988, water temperature measurements were taken manually during periodic sampling campaigns, or recorded using strip chart recorders from a limited number of depths during several years (1961-1963). As a consequence, information about the thermal state of the lake before 1988 was missing or infrequent for most of the time, and even after 1988 there are large gaps in the measured temperature data. Thus, the information available for Lake Erken made it a good study case for testing the methodological approach of this study. Here, we created a complete daily time step water temperature record for Lake Erken,

extending the information on its thermal structure further back in time until 1961, in order to provide a longer and more consistent picture of the changes that have occurred in the lake over the last five decades. The GOTM hydrodynamic model used here is driven by meteorological data collected from Uppsala University's field station at Lake Erken (http://www.ieg.uu.se/erken-laboratory/) and from nearby stations to create daily time-step water temperature profiles for the entire period 1961-2017. In this work we evaluated (1) the validity of using modelled temperature to reconstruct past water temperature of Lake Erken for the period 1961-2017, providing a valuable method that can be extended to other lakes, and (2) how water temperature and other metrics of lake stratification have changed over the study period. Finally, the creation of a reliable, consistent and complete 57-year dataset of daily water temperature profiles will be a valuable source of information for future research on Lake Erken that will help to better our understanding of many ecological processes that can be affected by changes in thermal conditions.

## 2. Methods

### 2.1 The lake

The lake investigated in this study was Lake Erken (59.4166 N, 18.2500 E) a mesotrophic lake located ~60 km North-East from Stockholm (Sweden) at an altitude of 10 m above the sea level. The lake covers an area of about 24 $km^2$ and its catchment area is relatively small (141 $km^2$), mainly covered by forest and with no major anthropic activities (Malmaeus et al., 2005). Lake Erken has a mean depth of 9 m and a maximum depth of 21 m while its water retention time is around 7 years (Blenckner et al., 2002). Little is the contribution of inflows on lake hydrodynamics (Pierson et al., 1992). The lake is always ice-covered in its entirety during winter between December-February to March-May (Blenckner et al., 2002; Persson and Jones, 2008) and is always stratified during summer months between May-June to August-September (Persson and Jones, 2008).

### 2.2 The model

The model used in this work is GOTM (General Ocean Turbulence Model). GOTM is an open source 1-dimensional physical model for hydrodynamic applications in natural waters, it simulates processes such as surface heat fluxes, surface mixed-layer dynamics and stratification processes. Detailed information about GOTM can be found in Burchard (2002) and on the website www.gotm.net. In this study, seven climatic parameters were used as input to the GOTM model over the study period (January 1[st], 1961 - October 31[st], 2017): wind speed (WS; m/s), air temperature (Air T: °C), relative humidity (RH; %), atmospheric pressure (Air P; hPa), cloud cover (CC; dimensionless value between 0-1), shortwave radiation (SWR; $W/m^2$) and precipitation (DP; mm/day). The model utilises six of these climatic parameters (excluding DP) at an hourly time step; DP is input on a daily time step. For the purpose of this study, the lake was considered to have a fixed water level equal to the long-term mean. This assumption was justified given the lakes long retention time and that the mean annual variation in lake level is only 48 cm. The GOTM model used for the simulations documented here did not have a fully functioning ice model, but instead cut off surface heat exchange when the simulated surface water temperature become negative. This provides a very simple way

to make continuous simulations that include freezing conditions that would normally lead to the formation of ice. Also, the model does not take into account the heat loss from lake sediments during ice-cover, which causes an increase in bottom water temperature. For this reason, the temperature profiles during winter (Dec.-Mar.) were not realistic and were excluded from model calibration. However, the onset and loss of stratification always falls between Apr 1[st] – Nov 30[th] (the period used for calibration), showing that the lack of a fully functioning ice model did not influence simulated estimates of the timing and duration of thermal stratification. A visual example that describe the mismatch between modelled and observed temperature in winter is available in the supplementary material (fig. S1-S2).

### 2.3 Data sources of meteorological parameters

Driving meteorological parameter were retrieved whenever possible from the Erken laboratory meteorological station (Malma islet; 59.8391 N, 18.6296 E, fig. 1a, letter A) and Svanberga meteorological station (59.8321 N, 18.6348 E, fig. 1a, letter B), about 800 m from the Erken laboratory meteorological station. The Svanberga meteorological station is managed by SMHI (Swedish Meteorological and Hydrological Institute) and data were downloaded from SMHI website (http://opendata-download-metobs.smhi.se/explore/). Meteorological data from neighboring meteorological stations were used when data from Erken or Svanberga were not available (fig. 1b). When data could not be retrieved from neighboring stations, missing data were replaced by linear interpolation. An overview of the number of meteorological data retrieved from different sources is given in table 1. A detailed description on the methodology used to retrieve these data is available in supplementary material.

### 2.4 Missing data replacement and missing data estimation with Artificial Neural Network analysis

To simulate Lake Erken water temperature at daily time step using GOTM, a continuous hourly record of meteorological forcing data was created by merging the data sources described above.  In the case of DP missing data were replaced by taking data from the closest stations to Lake Erken (see supplementary material). For CC that was only available from one station (Svenska Hogarna: 59.4445 N, 19.5059 E), missing data were replaced by linear interpolation. To make maximum use of data from surrounding stations we used an Artificial Neural Network fitting analysis (ANN *nftool*) to predict missing meteorological data at Erken (DP and CC excluded). The analysis was carried out using MATLAB version R2017b (MathWorks Inc. Natick, Massachusetts).

ANN algorithms were used to estimate the driving parameters during occasions when no local measurements were recorded at Erken meteorological station (WS, Air T and SWR) and Svanberga (Air P and RH). Input data were those collected from the nearest (less than 60 km away) meteorological stations to Lake Erken and Svanberga. Input data were retrieved from SMHI database, except for SWR data that were retrieved from measurements made at the Swedish Agricultural University (SLU) near Uppsala. The choice of the meteorological datasets to use as input data was based on two characteristics:

1.  Offsite datasets that have recorded data when data from Lake Erken or Svanberga were not available.
2.  Offsite and local dataset overlap for at least 8 years to get a reasonable number of data to perform ANN function fitting analysis that describes input-target relationship.

A detailed description of the Neural Network analysis is available in supplementary material.

## 2.5 Observed water temperature data and model calibration

The model was calibrated using measured profiles of averaged daily water temperature collected between April 1[st] - November 30[th] when the lake is usually ice-free. Observed data in the period 1961-1988 were collected manually during occasional

sampling campaigns, and from strip chart data recordings made at the Erken meterological station. Most of the observed temperature data were measured at 0.5, 5, 10 and 20 m depth. A much greater number of observed data were available for the period 1989-2017 when an automated floating station (59.84297 N, 18.635433 E, fig. 1, letter C) was deployed to collect water temperature data during the ice-free period. The floating station measured water temperature data every 0.5 meters, from 0.5 m to 15 m depth. Profiles were stored every 30 minutes and these were averaged to provide a daily mean profile.  Also between

1989-2017 water temperature was digitally recorded in a manner similar to the old strip chart recordings.  These measurements were made year-round from the 1, 3 and 15 m depths at the Erken meteorological station, and used for calibration at times the floating system was not deployed. The total number of observed water temperature data in Apr.-Nov. between 1961-2017 was 103454. The number of days with at least one single observed measurements was 6674 days between 1961-2017.

The program ACPy (Auto Calibration Python) was used to calibrate the model (webpage: www.bolding-

bruggeman.com/portfolio/acpy/). ACPy is a utility that eliminates the need for time-consuming manual calibration of hydrodynamic and water quality models.  This allows for more extensive testing and evaluation of model calibrations, ultimately providing more accurate and repeatable results. A set of model parameters was calibrated and adjusted within their feasible range (see table 2) in order to minimize the difference between the simulated and measured water temperature. Simulations were run between 1961 and 2017 but in order to obtain stable initial conditions the model was run for an additional

one year spin up using a copy of the 1961 data. In this way, 1961 data were both used as spin-up year prior to the calibration and then reused in the proceeding calibration.

In ACPy, a Differential Evolution algorithm (Storn and Price, 1997) is used to calculate a log likelihood function, which compares the modelled water temperature to the observed temperature.

The likelihood $\Lambda$ is defined in Eq. (1):

$$\Lambda = - \sum_i \frac{(x_{obs,i} - x_{mod,i})^2}{var_x} \tag{1}$$

where $x_{obs,i}$ is the observed temperature, $x_{mod,i}$ is the modeled temperature and $var_x$ is the variance between the modelled and observed temperature.  ACPy was set to run 10000 simulation during calibration in order to obtain a stable solution and get the optimal parameter set that minimizes the log likelihood function. Following the calibration, model fit was judged based on estimates of bias, mean absolute error (MAE), and the root mean squared error (RMSE).

For typical applications of models where the goal is to make simulations to future or otherwise different conditions that are not covered by the record of measured calibration data, it is appropriate to employ a split calibration and validation strategy.

However, in our case the goal was not to simulate outside of the period of available calibration data, but to use the model to provide a complete and consistent record over a period in which calibration data were available but incomplete. Therefore, we used here the entire record of measured temperature over the study period (1961-2017) to judge the validity of the calibration. This ensured that the calibration encompasses the widest possible range of variability and that our simulations within the calibration period would have the greatest degree of accuracy. The adjusted model parameters in this study are non-dimensional scaling factors that adjust the heat-flux, shortwave radiation and wind as well as the minimum turbulent kinetic energy and the e-folding depth for visible fraction of incoming radiation which are parameters that strongly influence the vertical distribution of light and temperature in the water column. Using ACPy all of these parameters were adjusted to minimize the difference between observed and modelled temperature. For the purpose of generating the long-term time series of temperature data we used the full calibration period (all seasons but winter) because it encompassed the full variation of conditions that occur over the entire simulation period. However, in addition to that, we also performed seasonal calibrations in order to validate that the model performed well in all seasons (winter excluded), especially when simulating the onset of stratification after the ice break-up.

The best set of parameters calculated from the ACPy full calibration period (table 2) were then used for the final simulation which produced the data analyzed in the remainder of this paper. The model fit results for the full period and seasonal calibration are shown in table 3. The comparison between observed and simulated water temperature and the error distribution after full calibration period are shown in figure 2. We also calculated the model performance at different depths (0.5, 5, 10 and 15 meters) after full calibration period. The results are shown in figure 3-4.

## 2.6 Statistical analysis

The entire statistical analysis was carried out using R Studio version 3.4.1 (R Studio Team 2016). We summarized the model temperature output by calculating a number of statistics that can qualify the ecological consequence of changes in thermal stratification using Lake Analyzer R Package (Winslow et al., 2018). The ecological implications of the changes of these metrics due to climate change are discussed in detail in the Discussion section. Lake Analyzer calculated volumetrically averaged epilimnetic (upper layer during stratification), hypolimnetic (lower layer during stratification) and whole-lake temperature, the thermocline depth (°C), Schmidt stability (J m$^{-2}$, Schmidt 1928, Idso 1973), and difference between surface and bottom temperature ($\Delta T$, °C). In addition, we calculated the length of the growing season for each year, defined as the number of days in which epilimnetic temperature exceeds 9 °C (Håkanson and Boulion, 2001).

In this study, the lake was considered stratified when the difference between surface temperature and bottom temperature ($\Delta T$) was greater than 1 °C (Woolway et al., 2014). The onset of stratification was considered to be the first day of the first period of 4 or more consecutive days in which $\Delta T > 1$ °C (Yang et al., 2016), and, in general, stratification events shorter than 4 days were also not considered when estimating duration and loss of stratification.

To assess if water temperature trends vary with seasons, the seasonal averages of the simulated water temperature were analyzed using the non-parametric Mann-Kendall test (Mann 1945, Kendall 1975). This test assesses whether a statistically

significant monotonic increase or decrease is occurring over time. The values of such trends were estimated using the non-parametric Sen's slope (Sen 1968), which is the median of all pairwise slopes of the considered data. Since the simulation stops in October 2017, autumn water temperature of that year were not taken into consideration in the data analysis. The Mann-Kendall and Sen's slope were also used to evaluate trends in average Schmidt stability during thermal stratification, thermocline depth, stratification duration, onset and termination of stratification and growing season length. Data autocorrelation was tested using *acf* and *pacf* function in RStudio. For autocorrelated data, the modified version of Mann-Kendall test proposed by Hamed and Rao (1998) was used instead of the traditional Mann-Kendall test, which does not account for autocorrelation. We used the Pettitt test (Pettitt 1979) to assess whether an abrupt change in annual mean air temperature occurred during the study period. Since Lake Erken is always ice-covered during winter and the GOTM model does not contain an ice-cover module, the simulated winter lake temperature were underestimated, especially in the bottom layers, where GOTM did not simulate the effect of heat flux from the sediment into the water. For this reason, trends in winter lake temperature were not analyzed in this study. However, the availability of manual observations of the timing of ice cover since 1941 (for 10 out of 68 years ice cover data are not available) for Lake Erken made it possible for us to test for trends in ice cover length during the interval 1941-2017, and to make comparison with the other simulated lake metrics. Mann-Kendall test, Sen's slope and Pettitt test were therefore used to analyse the record of ice cover length obtained from observational data. A synthesis of the statistical analysis results is reported in table 4.

## 3. Results

### 3.1 Model performance

The GOTM model was able to accurately reconstruct water temperature of Lake Erken in the time interval 1961-2017. Overall, the calibrated model showed a RMSE of 1.089 ℃, a MAE of 0.753 ℃ (Table 3, fig. 2). The modelled temperature showed a slightly cold temperature bias (-0.047 ℃). The comparison between observed and modelled temperature at specific depths (0.5, 5, 10 and 15 m) showed a good model performance throughout the entire water column. At 0.5 m depth (fig. 3a), the calculated RMSE was 0.827 ℃ and the MAE was 0.614 ℃. The modelled temperature at 0.5 m were slightly warmer than the observed water temperature, since the measured bias was 0.086 ℃. At 5 m depth (fig. 3b), RMSE and MSE were very similar to the ones calculated at 0.5 m with a value of 0.840 ℃ and 0.618 ℃ respectively. A slightly colder temperature bias was found (-0.004 ℃). The comparison of modelled and observed temperature at 10 m depth showed a RMSE of 1.187 ℃, a MAE of 0.811 ℃ and a temperature bias of 0.003 ℃ (fig. 4a). At 15 m depth, the RMSE was 1.155 ℃, the MAE was 0.803 ℃ and the temperature bias was -0.137 ℃ (fig. 4b).

### 3.2 Reconstructed thermal structure

The Pettitt test showed that a significant abrupt change in annual mean air temperature occurred in 1988 (p < 0.001). Therefore, in addition to checking for trends in lake thermal structure over the entire simulation period we also evaluated the possibility

of trends occurring over the subinterval 1961-1988 and 1989-2017, and we tested whether a more rapid change in water temperature is occurring after 1988, following a step-change in annual mean temperature. The Mann-Kendall test showed that during summer (Jun.-Aug.) a significant increase in whole-lake and epilimnetic temperature of 0.239 °C/decade (p-value < 0.001, fig. 5a) and of 0.351 °C/decade (p-value < 0.001, fig. 5b) respectively occurred over the entire study period (1961-

2017), but not when the trend analysis was performed in the sub-intervals. Similarly, a slightly increasing trend was also detected for hypolimnetic spring temperature (0.148 °C/decade, p-value < 0.05, fig. 5c) over the entire study period but not in the sub-intervals. No other significant trends were detected over the entire simulation period or over the sub-interval 1961-1988. In contrast, the results suggest that Lake Erken did change more rapidly since 1989. Significant positive trends were detected from 1989 onwards during both the spring and autumn. Since 1989, spring (Apr. – May) whole-lake temperature

showed an average increasing trend of 0.404 °C/decade (p-value < 0.05, fig. 5a) and epilimnetic temperature an average increasing trend of 0.444 °C/decade (p-value < 0.05, fig. 5b). The same pattern is shown during autumn months (September-November) with no trends detected in the sub-interval 1961-1987, while significant increasing trends were detected in the sub-interval 1988-2016 for whole-lake (0.789 °C/decade, p-value < 0.001, fig. 5a), epilimnetic (0.792 °C/decade, p-value < 0.001, fig. 5b) and hypolimnetic temperature (0.816 °C/decade, p-value < 0.01, fig. 5c).

Other metrics of thermal statification showed long-term trends that were significant over the entire simulation period. The length of the growing season showed a positive significant increase, which, on average, was of 3.793 days/decade (p-value < 0.001, fig. 6) in the interval 1961-2017. With regards of thermal stability, the trend analysis of Schmidt stability revealed that more energy is required to mix the lake during stratified conditions in recent years if compared to the first years of the study period (5.365 Jm$^{-2}$/decade, p-value < 0.01, fig. 7). This greater stability also corresponded with a longer duration of

stratification. From 1961, the duration of lake stratification increased, on average, by 7.297 days/decade (p-value < 0.001, fig. 8a). The longer period of stratification is the result of both an earlier onset of thermal stratification, which now occurs on average ~16 days earlier since 1961 (-2.903 days/decade, p-value < 0.01, fig. 8b) and a later loss of thermal stratification that now is on average delayed by ~26 days (4.583 days/decade, p-value < 0.001, fig. 8c). The difference between surface and bottom temperature is often used as a simple indicator of thermal stratification. Its mean annual value during the stratified

period increased significantly over time, increasing, on average, by 0.253 °C/ decade (p-value < 0.05, fig. 9). Mean annual thermocline depth during lake stratification period shows a significant decrease over the entire study period, with an average decrease of ~1.345 m since 1961 (-0.236 °C/decade, p-value < 0.01, fig. 10).

Regarding ice cover duration, the Pettitt test showed that an abrupt change in ice cover duration occurred in 1988 in the interval 1941-2017. Therefore, similarly to water temperature analysis, trend tests were performed in two sub-interval, 1941-1988 and

1989-2017. Trend in ice cover length did not significantly change within the sub-intervals. However, a significant decrease in ice cover length was detected when trend analysis were performed on the entire interval (-7.343 days/decade, p < 0.001, fig. 11).

## 4. Discussion

The model used in this study revealed a good match between observed and simulated water temperature during the entire study period 1961-2017 (RMSE = 1.089 ℃, MAE = 0.753 ℃, bias = -0.047). In particular, the GOTM model was able to reproduce past water temperature with a high level of accuracy not only when meteorological driving data were available from the Erken
meteorological station (1988-2017), but also during the period 1961-1988, when most of the meteorological data were estimated using Artificial Neural Network Analysis. Indeed, the model was able to well describe water temperature during the three-year period 1961-1963, when it was possible to compare frequent water temperature measurements recorded from several depths by strip chart recorders (fig. 3-4).

The model performance was very similar at 0.5 m and 5 m depth, with a RMSE of 0.827 ℃ and 0.840 ℃ respectively. Slightly
greater errors were found at 10 m and 15 m depth with a RMSE of 1.187 ℃ and 1.155 ℃ respectively. Since Erken is subjected to internal seiche movements, it is likely that higher errors found at deeper points (especially when a thermocline is present) between modelled and observed temperature could be at least partially explained by the limitation of a 1D model like GOTM to describe seiche movements. On seasonal basis, the model performed well in all the season (winter is excluded from the calibration). In spring (Apr.-May), the model showed a RMSE of 0.952 ℃, a MAE of 0.721 ℃ and a very low temperature
bias (-0.008 ℃). These values revealed that, despite the lack of a fully functioning ice-module in the GOTM version used here, the model performed very well during a period in which ice-cover could still occur in some years at Lake Erken. The good model performance during spring also adds confidence to the onset of stratification calculated from the modelled water temperature profiles, which often starts in April or May. In summer (Jun.-Aug.), the model showed slightly higher errors (RMSE = 1.240 ℃, MAE = 0.903 ℃ and bias = -0.027 ℃). Lake Erken is always stratified during summer and, similarly to
higher errors found at deeper points, higher errors of the model in this season could be related to seiche movements around the thermocline that are hard to predict with a 1D model. This is corroborated by the fact that in autumn (Sep.-Nov.), when the lake is fully mixed, the model performed better (RMSE = 0.530 ℃, MAE = 0.361 ℃, bias = -0.005 ℃).

The Pettitt test showed that an abrupt change in air temperature occurred in 1988, which is consistent with the results of Temnerud and Weyhenmeyer (2008), who found that the most abrupt changes in air temperature (interval 1961-2005) occurred
in 1988 and 1989 across different sites in Sweden. The similarity of these findings with the present work demonstrates the consistency of air temperature data used to drive GOTM model to that previously evaluated by Temnerud and Weyhenmeyer (2008), and also supports our finding that trends in water temperature were strongest during the period 1989-2017, and that this interval plays the most important role in defining the trends water temperature warming. Since the majority of the increasing trends were detected only in this sub-interval, it is apparent that most of the increase in Erken water temperature
has occurred during the last three decades rather than during the entire study period. Overall, autumn is the season that showed the highest increase in water temperature between 1989-2017 (whole-lake: 0.789 °C/decade, epilimnion: 0.792 °C/decade, hypolimnion: 0.816 °C/decade). A lesser trend was detected during spring between 1989-2017 for whole-lake and epilimnetic temperature (0.404 °C/decade, and 0.444 °C/decade respectively). Summer whole-lake and epilimentic temperature showed a

constantly increasing trend throughout the entire study period, with a significant increase since 1961 (0.239 °C/decade and 0.351 °C/decade respectively), but no significant increase in the sub-intervals, suggesting that the first and most persistent effects of global warming have occurred during summer. Otherwise, the more recent and more significant trends are most apparent in spring and autumn. These results also showed that while epilimnetic temperature increased in each season,

hypolimnetic temperature showed a significant increase in autumn between 1989-2017 (0.816 °C/decade) in autumn and in spring, even though the trend detected in this season is pretty low (0.148 °C/decade over the interval 1961-2017), while no significant trends were detected in summer. The marked trend detected in autumn could results from the entrainment of warmer epilimnetic waters into the hypolimnion as the seasonal thermocline deepens. In general, however, the lower and fewer increasing trends detected in the hypolimnion compared to epilimnion and whole-lake temperature could be related to the fact

that hypolimnetic temperature is less affected by meteorological variability than epilimnetic temperature (Adrian et al., 2009). A large-scale study carried out by O'Reilly et al., (2015) found that a mean trend of increasing summer surface water temperature (0.34 °C/decade) was detected in 235 worldwide between 1985-2009, with trends from individual lakes ranging from -0.7 - +1.3 °C /decade. The study reports a surface water temperature trend for Lake Erken derived from measured data of 0.61 °C/decade (see O'Reilly et al., 2015, supporting information), while in the present work the trend of the surface summer

modelled temperature calculated over the same time period (1985-2009) is somewhat greater: 1.145 °C/decade. The trend detected by O'Reilly et al. (2015) is calculated using a dataset with temporal gaps in the summer water temperature record (14 years with recorded data between 1985-2009), while in the present work the trend was calculated using a complete long-term dataset of simulated summer surface water temperatures. Furthermore, the same article indicates that water temperature trends detected for lakes with data gaps might have been underestimated, suggesting that the trend detected in the present study for

Lake Erken could be more accurate. This illustrates the value of using more complete and consistent modelled data to calculate trends, and also indicates that the lake is warming at a rate near the global maximum. The rapid rate of warming estimated from our work is also consistent with the conclusions of O'Reilly et al. (2015), that lakes located in Northern Europe are warming more rapidly than the global average, and also of Kraemer et al. (2017), that lakes at high latitudes are warming faster than tropical lakes. Temperature trends obtained in the present study are consistent with these findings.

A prolonged duration of high surface water temperature and an increase of epilimnetic temperature can have impacts on lake mixing dynamics leading to a higher thermal stability (Jankowski et al. 2006; Butcher et al., 2015). Such increases in water temperature can explain why Schmidt stability has also increased over the period 1961-2017 (5.365 Jm$^{-2}$/decade) and why the duration of stratification has also increased by about 40 days since 1961, shifting both the onset and the loss of the stratification. Compared to 1961, the present onset of thermal stratification occurs on average 16 days earlier. However, the higher thermal

stability has even a greater effect on the loss of stratification. From the simulated temperature, the end of the stratification now occurs on average 26 days later if compared with the 1960s. Very similar results were reported by Arvola et al. (2009) who used less frequently measured temperature data to estimate that the loss of stratification in Lake Erken was delayed by almost one month since the 1960, further verifying the reliability of the model based approach used in this study for detecting such variation. However, Arvola et al. (2009) did not detect the trends in the onset and duration of stratification or lake warming

that were detected here. That these trends are now further detected shows the value of using model simulations to provide long-term consistent temperature records that are more amenable to trend analysis.

Higher surface temperature also increased the difference between surface and bottom temperature over the period 1961-2017 (0.253 ºC/decade) thereby increasing the gradient of the thermocline, while reducing the mean thermocline depth (-0.236 ºC/decade). This could in part be due to a lower wind speed in recent times. Trend analysis of mean annual wind speed have revealed that there is a significant decreasing trend in wind speed over the study period. Since 1961, the wind speed has decreased on average by 0.775 m/s (Sen's slope = -0.136 ms$^{-1}$/decade, $p < 0.001$). Another possible explanation to a shallower thermocline could be related to a reduction of heat fluxes from water to air, which might have weakened the convective mixing of the upper layers (Monismith and MacIntyre 2009). However, heat fluxes have not been analysed in this study, and further research is needed to better understand the causes behind reductions in thermocline depth.

A known limitation of the GOTM model version used in this work is the lack of an ice-module to simulate the onset, the loss and the duration of the lake ice-cover. Simulated and observed temperature close to the surface, are similar throughout the entire year. However, during winter months (Dec.-Mar.) the model does not take into account the heat loss from sediment during ice-cover, which cause an increase in bottom water temperature. Despite the lack of simulated ice cover, it was possible to analyse the duration of ice cover thanks to a yearly observations of the onset and loss of ice cover made at Lake Erken since 1941. Ice cover duration is decreased since 1941 of 7.343 days/decade. A step-change in ice cover duration was detected in 1988, consistent with the step change in air temperature detected in the present study and by Temnerud and Weyhenmeyer (2008). When trend analysis was performed on the two sub-intervals 1941-1988 and 1989-2017 no significant trends in ice cover duration were detected. However, the ice cover duration showed a greater interannual variability in the sub-interval 1989-2017 compared to the sub-interval 1941-1988. Within the sub-interval 1941-2017, the variability in ice cover duration ranges from a minimum of 68 days in 1961 to a maximum of 168 days in 1958, while in the sub-interval 1989-2017 the ice cover duration ranges from 12 days in 2008 to 142 days in 2011. Such increase in variability of ice cover duration could be related to warming of climatic conditions (Magnuson et al., 2000). Sadro et al. (2019) found that decline of snowpack in mountain lakes in Sierra Nevada (California) causes a warming response in lake temperature. Since Lake Erken and most lake in Scandinavia are always ice covered during winter and snowfall occurs every year, understanding the dynamics of snowfall and ice cover phenology could be of extreme importance to better understand thermal response of lakes to climate change.

Changes in lake water temperature and stratification patterns can have a broad influence on many aspects of lake ecosystems, both biotic and abiotic. For example, a longer duration of thermal stratification could lead to a depletion in hypolimnetic oxygen (Jankowski et al., 2006, Butcher et al., 2015), potentially reducing the natural range of lacustrine fish (Jones et al., 2008) or otherwise influence the vertical distribution of living organisms (Woolway et al., 2014). Moreover, an earlier onset of thermal stratification and warmer lake temperature could change the seasonal dynamics of phytoplankton (Thackeray et al., 2008). A previous model simulation conducted by Blenckner et al. (2002) on the ecology of Lake Erken concluded that warmer water temperature and changes in mixing dynamics due to climate change are likely to boost nutrient concentrations and phytoplankton production, with consequences for the entire lake ecosystem in the coming decades. Given the relative long

retention time of lake Erken (7 years), the importance of internal phosphorus loading due to changes in thermal stratification could make the lake more susceptible to climate change than other Swedish lakes with shorter retention times and higher levels of external nutrient loading (Malmaeus et al., 2005). Another potential impact could be an increase of carbon emission from the lake, since a recent study has shown that an increase in nutrient concentration coupled with a rise in water temperature can

have a positive and synergistic effect on methane ebullition (Davidson et al., 2018), and in a warmer world not only methane, but also $CO_2$ emissions from boreal lakes are likely to increase (Weyhenmeyer et al., 2015). Finally, a general indicator of the effects of warmer conditions on the biological dynamics of Lake Erken is the growing season indicator of Håkanson and Boulion (2001). Our simulations showed a significant increase in the number of days in which epilimnetic temperature was greater than the suggested 9 ° C threshold during the 1961-2017 study period (3.793 days/decade).

**5. Conclusion**

The present study has shown that the use of the GOTM model to reconstruct the past 57-years of thermal condition of Lake Erken provided a valuable source of information that could be used to detect changes in its thermal structure. This methodology can be extended to other lakes that have incomplete records of water temperature data. The use of local meteorological data to drive model simulations such as those demonstrated here can be used to extend water temperature records further back in time

or fill data gaps where they exist. This work also shows that water temperature has been rising faster in the last three decades compared to the previous decades both in the epilimnion and hypolimnion and that other metrics describing thermal stratification have changed over the entire 57-year study period.

It is likely that increasing water temperature will cause many secondary effects with serious and to some extent unpredictable repercussions on lake ecosystems. This work can be seen as a baseline for future research on Lake Erken that involve climate-

related investigations. A further step towards a better understanding of how the lake ecosystems will respond to climate change is to couple a biogeochemical model with the physical model GOTM using FABM - Framework for Aquatic Biogeochemical Models (Bruggemann and Bolding, 2014). Then parameters such as chlorophyll, nutrient and dissolved organic carbon concentrations can be simulated and analyzed. The coupling of physical and biogeochemical models could, therefore, be a valuable tool to facilitate the mitigation of detrimental effects of a warmer world on lake ecosystems.

**Data and code availability**

The model and calibration configuration, the input data used to run the GOTM model, the output data and the water temperature used to calibrate the model are available on Hydroshare (doi:10.4211/hs.54375615d258461086125d5fc85a4c32). Matlab codes, R codes and all the datasets produced during this study are available upon request from the corresponding author.

**Author contributions**

DCP designed the study, SM and AIA processed meteorological data used in this study, performed model simulation and data analysis. SM wrote the first draft of the paper with contribution of all the co-authors. SM wrote the final version of the paper with contribution of DCP.

**Competing interests**

The authors declare no competing interests.

**Acknowledgements**

The authors are grateful to Thomas Carlund from SMHI for providing some of the meteorological and water temperature data used in this study and for his assistance with interpretation of solar radiation data.

The authors would like to thank the EU and FORMAS for funding, in the frame of the collaborative international Consortium PROGNOS financed under the ERA-NET WaterWorks2014 Cofunded Call. This ERA-NET is an integral part of the 2015 Joint Activities developed by the Water Challenges for a Changing World Joint Programme Initiative (Water JPI).

We acknowledge funding from the project WATExR which is part of ERA4CS, an ERA-NET initiated by JPI Climate, and funded by MINECO (ES),FORMAS (SE), BMBF (DE), EPA (IE), RCN (NO), and IFD (DK),with co-funding by the European Union (Grant 690462) and FORMAS grant 2017-01738.

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

**Tables**

**Table 1: Number of data points retrieved from difference sources of the driving meteorological parameters for GOTM simulation**

| Meteorological parameter | No. of data retrieved from Erken or Svanberga meteorological station | No. of data retrieved from neighbouring meteorological stations | No. of interpolated data | Total data |
|---|---|---|---|---|
| WS | 282389 | 188567 | 28700 | 499656 |
| Air T | 235250 | 234982 | 29424 | 499656 |
| RH | 191678 | 294092 | 13886 | 499656 |
| Air P | 194881 | 259563 | 45212 | 499656 |
| SWR | 398129 | 101520 | 7 | 499656 |
| CC | - | 157948 | 341708 | 499656 |
| DP | 10016 | 10803 | - | 20819 |

5    **Table 2: Best parameter set from ACPy over the entire calibration period (Apr.-Nov.) between 1961-2017**

| Model parameter | Calibrated factor | Feasible range |
|---|---|---|
| Heat-flux factor | 0.863 | 0.5-1.5 |
| Short- wave radiation factor | 0.971 | 0.8-1.2 |
| Wind factor | 1.287 | 0.5-2.0 |
| Minimum turbulent kinetic energy | $1.649e^{-6}$ | $1.4e^{-7}$-$1.0e^{-5}$ |
| e-folding depth for visible fraction | 2.637 | 0.5-3.5 |

**Table 3: Model performance over the entire calibration period (Apr.-Nov.) and different seasons (spring, summer, autumn)**

| Calibration interval | Model statistics | Value |
|---|---|---|
| Apr.-Nov. (full period) | ln Likelihood | -60469.700 |
| | Bias (°C) | -0.047 |
| | MAE (°C) | 0.753 |
| | RMSE (°C) | 1.089 |
| Apr.-May (spring) | ln Likelihood | -7782.650 |
| | Bias (°C) | -0.008 |
| | MAE (°C) | 0.721 |
| | RMSE (°C) | 0.952 |
| Jun.-Aug. (summer) | ln Likelihood | -39818.975 |
| | Bias (°C) | -0.027 |
| | MAE (°C) | 0.903 |
| | RMSE (°C) | 1.240 |
| Sep.-Nov. (autumn) | ln Likelihood | 4122.193 |
| | Bias (°C) | -0.005 |
| | MAE (°C) | 0.361 |
| | RMSE (°C) | 0.530 |

**Table 4: Trend analysis results of the investigated lake metrics**

| Lake metrics | Time interval | Mann-Kendall τ | Sen's slope | Sen's Slope 95 % CI | P-value |
|---|---|---|---|---|---|
| Whole-lake spring | 1961-1988 | - | - | - | > 0.05 |
| | 1989-2017 | 0.305 | 0.404 ℃/decade | [0.076-0.827] | < 0.05 |
| Whole-lake summer | 1961-2017 | 0.308 | 0.239 ℃/decade | [0.119-0.381] | < 0.001 |
| | 1961-1988 | - | - | - | > 0.05 |
| | 1989-2017 | - | - | - | > 0.05 |
| Whole-lake autumn | 1961-1988 | - | - | - | > 0.05 |
| | 1989-2017 | 0.444 | 0.789 ℃/decade | [0.265-1.273] | < 0.001 |
| Epilimnion spring | 1961-1988 | - | - | - | > 0.05 |
| | 1989-2017 | 0.296 | 0.444 ℃/decade | [0.062-0.932] | < 0.05 |
| Epilimnion summer | 1961-2017 | 0.326 | 0.351 ℃/decade | [0.164-0.540] | < 0.001 |
| | 1961-1988 | - | - | - | > 0.05 |
| | 1989-2017 | - | - | - | > 0.05 |
| Epilimnion autumn | 1961-1988 | - | - | - | > 0.05 |
| | 1989-2017 | 0.455 | 0.792 ℃/decade | [0.248-1.262] | < 0.001 |
| Hypolimnion spring | 1961-2017 | 0.187 | 0.148 ℃/decade | [0.007-0.294] | < 0.05 |
| | 1961-1988 | - | - | - | > 0.05 |
| | 1989-2017 | - | - | - | > 0.05 |
| Hypolimnion summer | 1961-2017 | - | - | - | > 0.05 |
| Hypolimnion autumn | 1961-1988 | - | - | - | > 0.05 |
| | 1989-2017 | 0.392 | 0.816 ℃/decade | [0.262-1.323] | < 0.01 |
| Growing season (epi T > 9 ℃)* | 1961-2017 | 0.380 | 3.793 days/decade | [2.222; 5.319] | < 0.001 |
| Schmidt stability | 1961-2017 | 0.256 | 5.365 Jm$^{-2}$/decade | [1.900; 9.023] | < 0.01 |
| Stratification duration* | 1961-2017 | 0.420 | 7.297 days/decade | [4.667-10.500] | < 0.001 |
| Onset of stratification | 1961-2017 | -0.266 | -2.903 days/decade | [-4.314; -0.889] | < 0.01 |
| End of stratification* | 1961-2017 | 0.397 | 4.583 days/decade | [2.593; 6.250] | < 0.001 |
| $T_{surface}$-$T_{bottom}$ | 1961-2017 | 0.212 | 0.253 ℃/decade | [0.048; 0.464] | < 0.05 |
| Thermocline depth | 1961-2017 | -0.268 | -0.236 m/decade | [-0.380; -0.074] | < 0.01 |
| Ice cover length* | 1941-2017 | -0.307 | -7.343 days/decade | [-11.364; -3.438] | < 0.001 |

*Autocorrelated data. The dataset has been analyzed using the Modified Mann-Kendall test (Hamed and Rao, 1998)

**Figures**

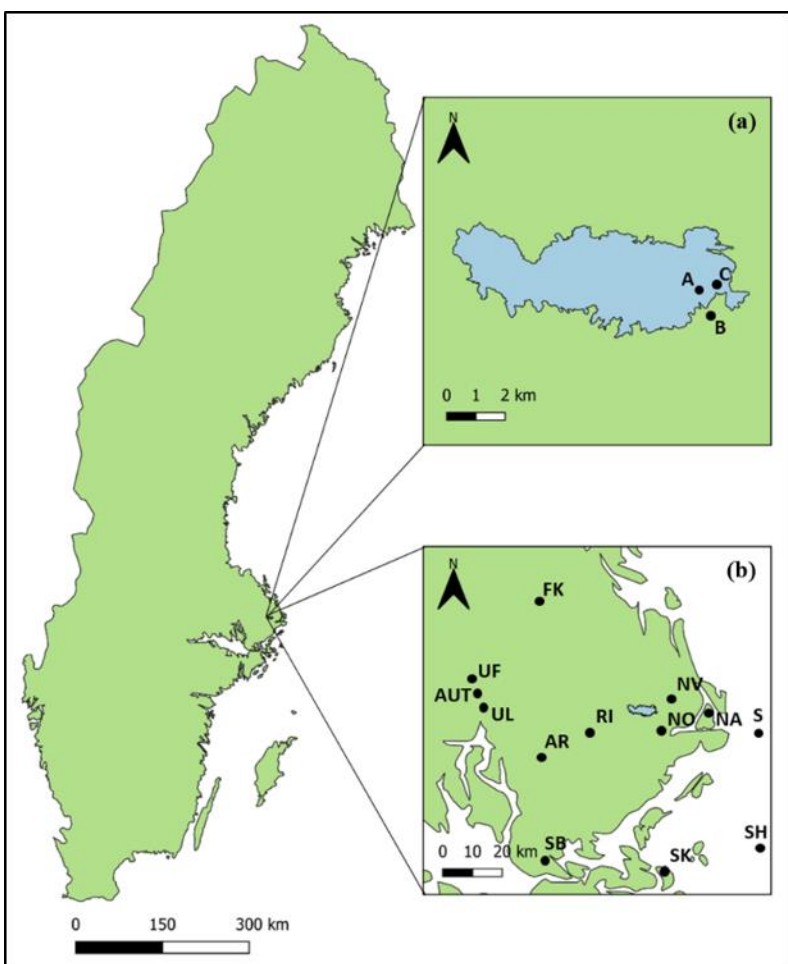

Figure 1: (a) Location of meteorological and floating stations within Lake Erken basin and catchment area. Letter A shows the position of Erken laboratory meteorological station (Malma islet, 59.83909 N, 18.62956 E), letter B identifies Svanberga SMHI weather station (59.8321 N, 18.6348 E), letter C represents the position of the floating station that records water temperature data. (b) Location of SMHI (Swedish Meteorological and Hydrological Institute) and SLU (Swedish Agricultural University) meteorological stations from which input data have been retrieved to run the model. SMHI stations: AR = Arlanda (59.6557 N, 17.9462 E), AUT = Uppsala AUT (59.8586 N, 17.6253 E), FK = Films Kirkby (60.2363 N, 17.9078 E), NA = Norrveda (59.8298 N, 18.9524 E), NO = Norrtälje (59.7506 N, 18.7091 E), NV = Norrtälje-Vasby (59.8524 N, 18.7296 E), RI = Rimbo (59.7487 N, 18.3535 E), S = Söderarm (59.7538 N, 19.4089 E), SB = Stockholm-Bromma (59.3537 N, 17.9513 E), SH = Svenska Hogarna (59.4445 N, 19.5059 E), SK = Skarpö A (59.3455 N, 18.7406 E), UF = Uppsala Flygplats (59.8953 N, 17.5935 E). SLU station: UL = Ultuna (59.8175 N, 17.6536 E).

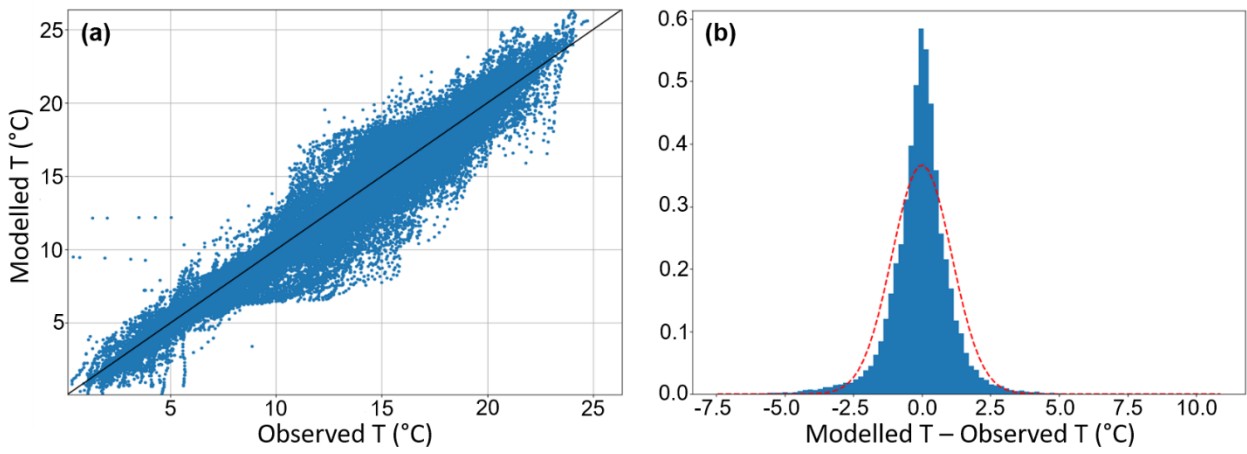

**Figure 2: (a) Comparison between observed water temperature and simulated water temperature of Lake Erken in the interval 1961-2017 (correlation = 0.972) and (b) error distribution between modelled water temperature and observed water temperature (Model-Observation) retrieved from ACPy calibration (panel b). RMSE = 1.089, MAE = 0.7529, bias = -0.04707.**

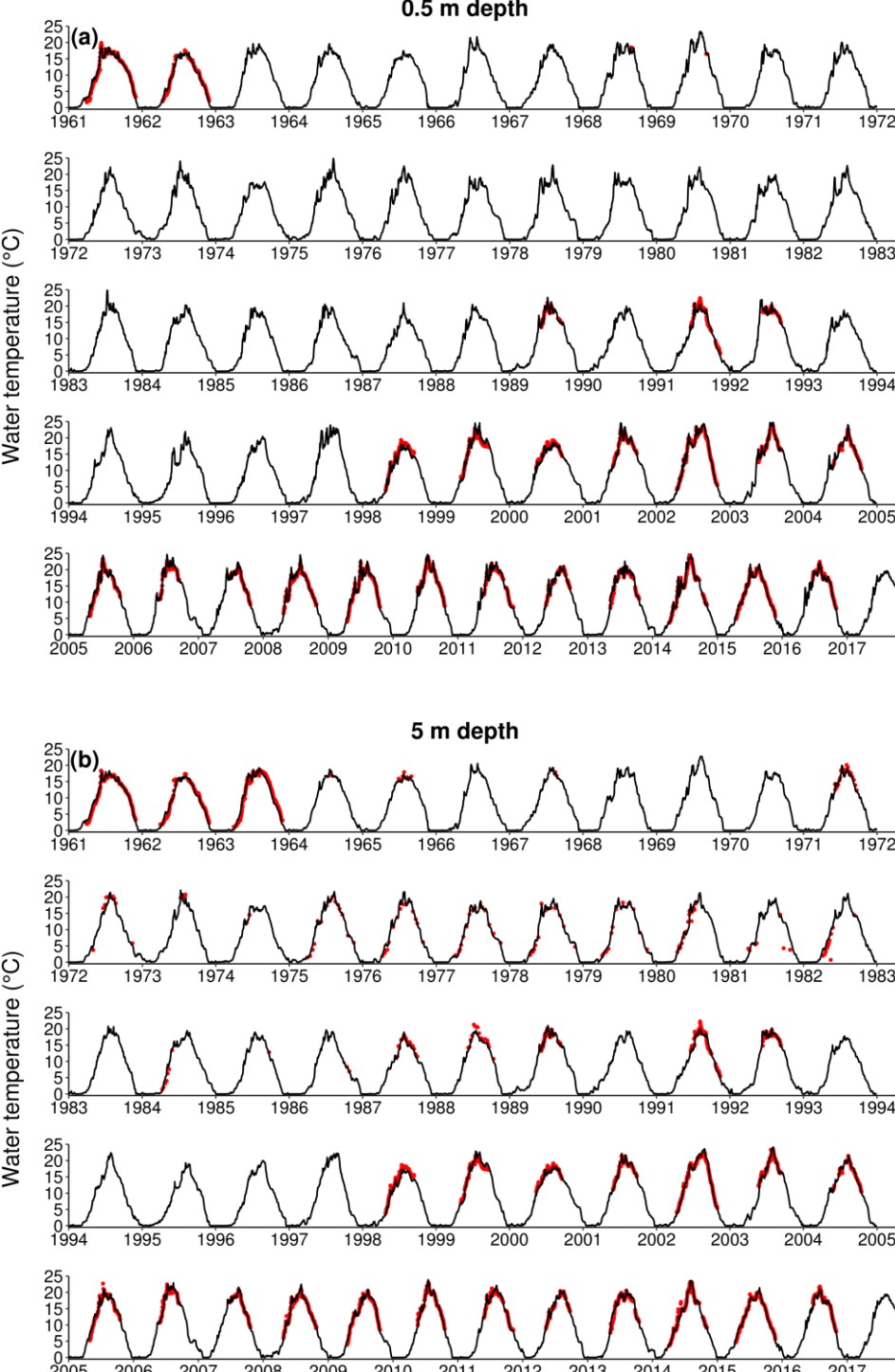

**Figure 3: Comparison between Erken modeled (black line) and observed daily temperature (red dots) at 0.5m (a), and 5m (b) depth (0.5m depth: RMSE = 0.827 ºC, MAE = 0.614 ºC, bias = 0.086 ºC; 5m depth: RMSE = 0.840 ºC, MAE = 0.618 ºC, bias = -0.004 ºC).**

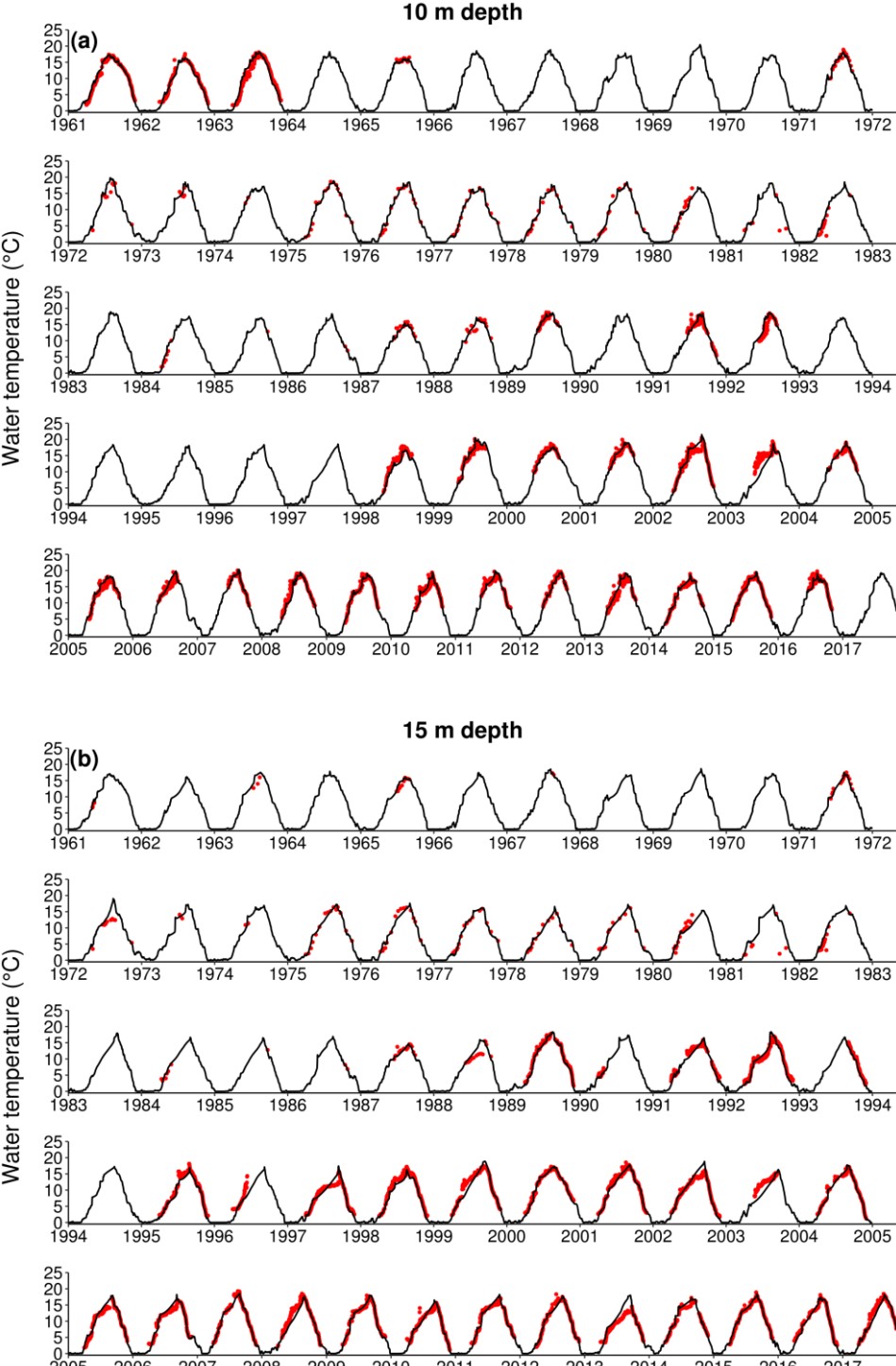

**Figure 4: Comparison between Erken modeled (black line) and observed daily temperature (red dots) at 0.5m (a), and 5m (b) depth (10m depth: RMSE = 1.187 ºC, MAE = 0.811 ºC, bias = 0.003 ºC; 15m depth: RMSE = 1.155 ºC, MAE = 0.803 ºC, bias = -0.137 ºC).**

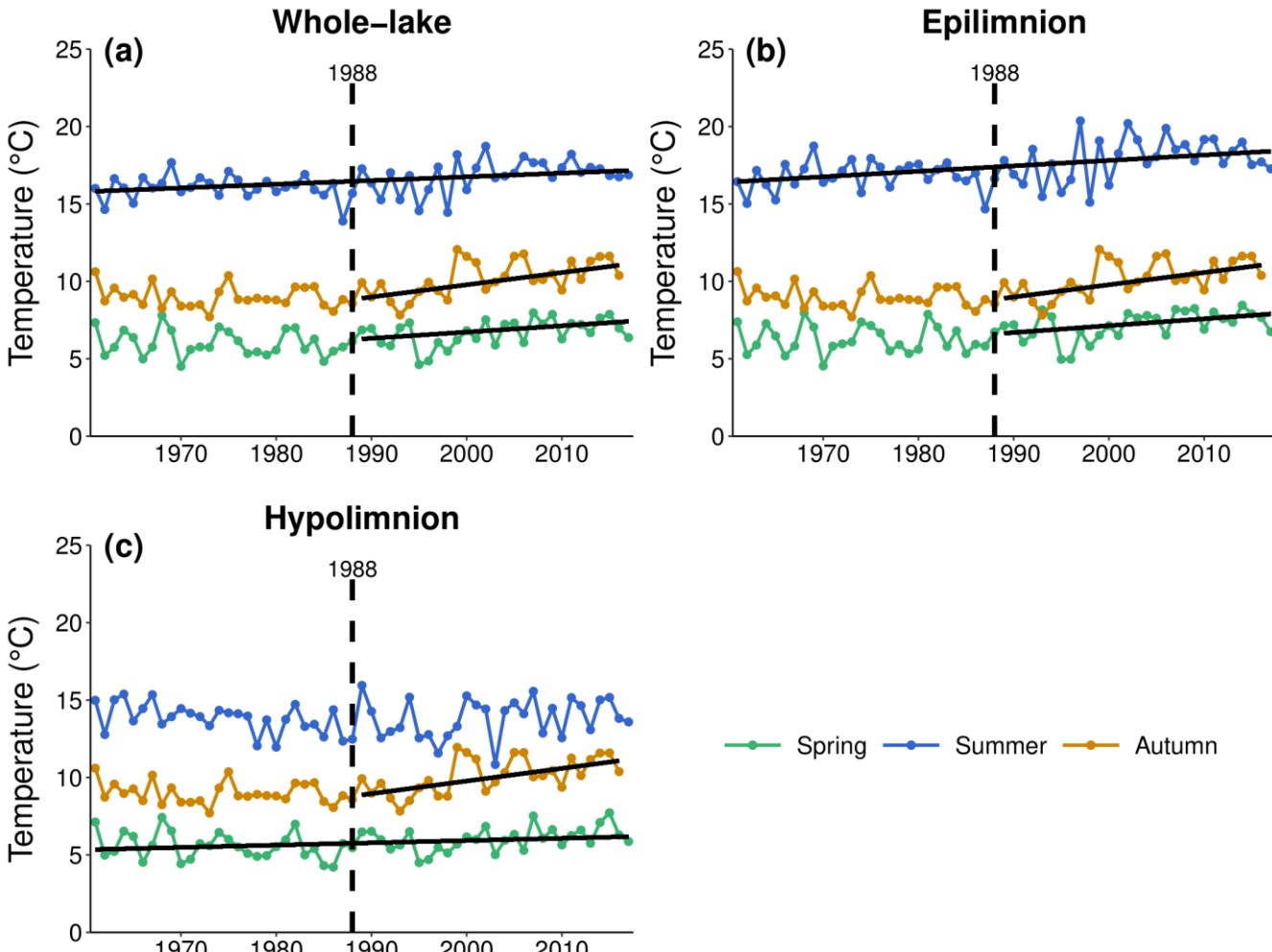

**Figure 5: Mean seasonal water temperature and relative trends for whole lake (a), epilimnion (b) and hypolimnion (c) between 1961-2017. Besides performing trend analysis over the entire study period, the Mann-Kendall analysis was performed in two sub-intervals (1961-1988 and 1989-2017) that are divided by the dashed line. Only significant trend lines are displayed. Whole-lake temperature significant trends: spring = 0.404 °C/decade, summer = 0.239 °C/decade, autumn = 0.789 °C/decade. Epilimnetic temperature significant trends: spring = 0.444 °C/decade, summer = 0.351 °C/decade, autumn = 0.792 °C/decade. Hypolimnetic temperature significant trends: spring = 0.148 °C/decade, autumn = 0.816 °C/decade.**

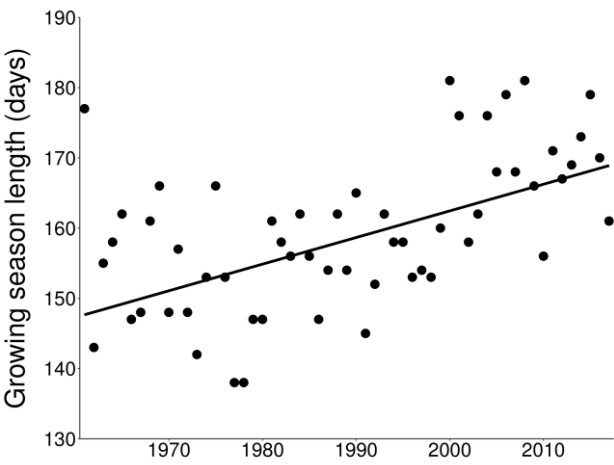

**Figure 6: Growing season length calculated from simulated water temperature between 1961-2017 (Sen's slope: 3.793 days/decade, p < 0.001).**

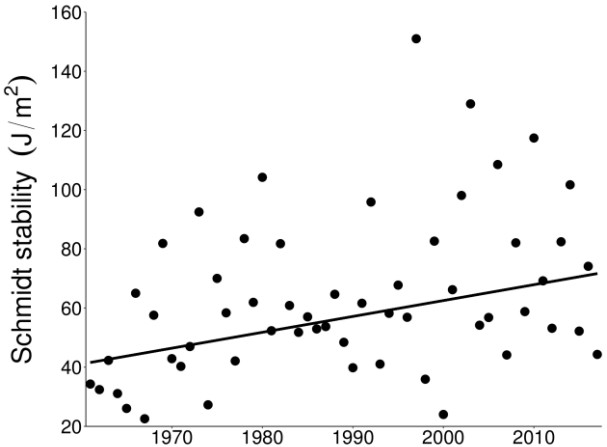

**Figure 7: Schmidt stability calculated from simulated water temperature between 1961-2017 (Sen's slope: 5.365 Jm$^{-2}$/decade, p < 0.01).**

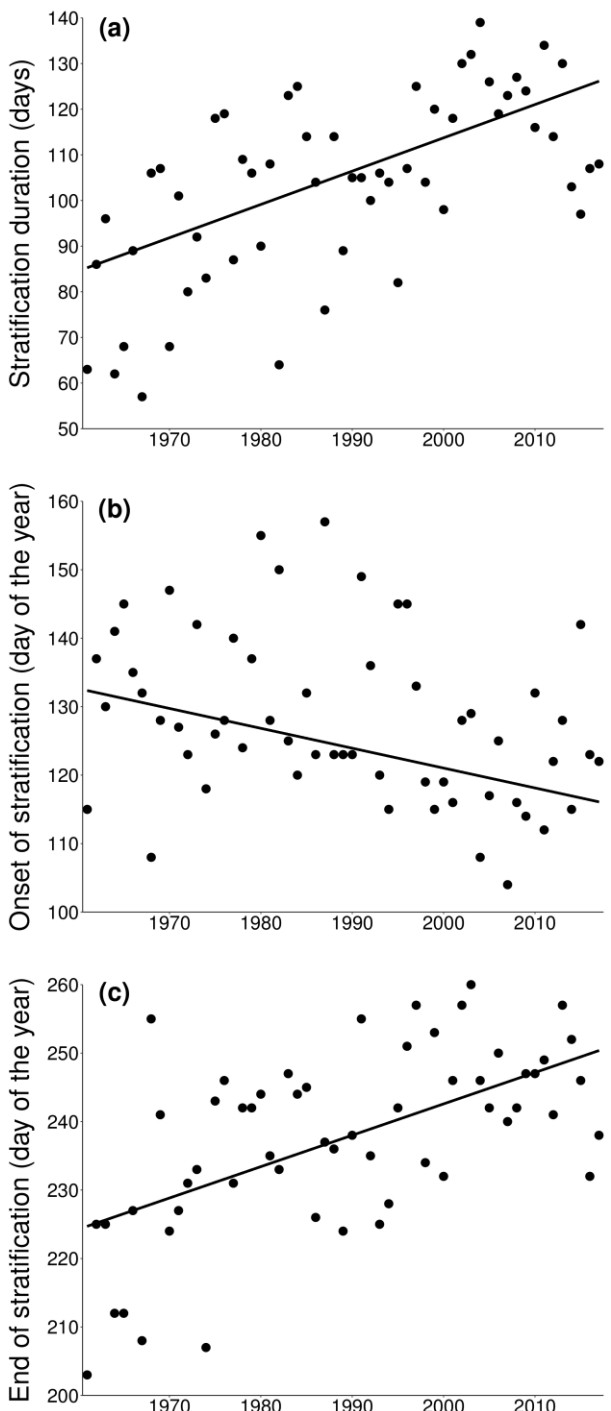

**Figure 8: Reconstructed stratification duration (a), onset (b) and termination (c) of Lake Erken between 1961-2017. Stratification duration trend: 7.297 days/decade (p < 0.001). Onset of stratification trend: -2.903 days/decade (p < 0.01). End of stratification trend: 4.583 days/decade (p < 0.001).**

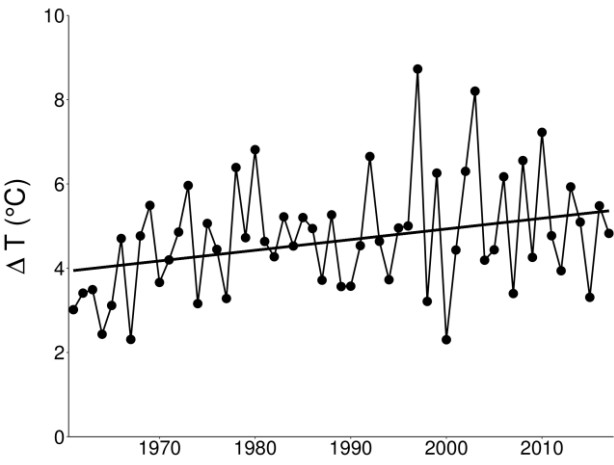

**Figure 9: Surface-Bottom modeled temperature difference (ΔT) during stratification between 1961-2017 (Sen's slope: 0.253 ºC/decade, p < 0.05).**

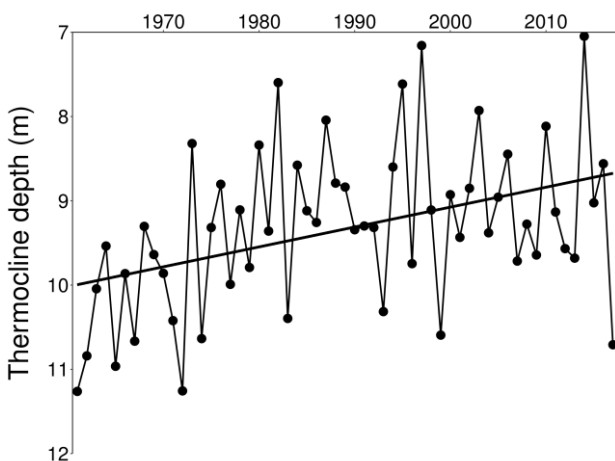

**Figure 10: Mean annual thermocline depth between 1961-2017 (Sens's slope: -0.236 m/decade, p < 0.01).**

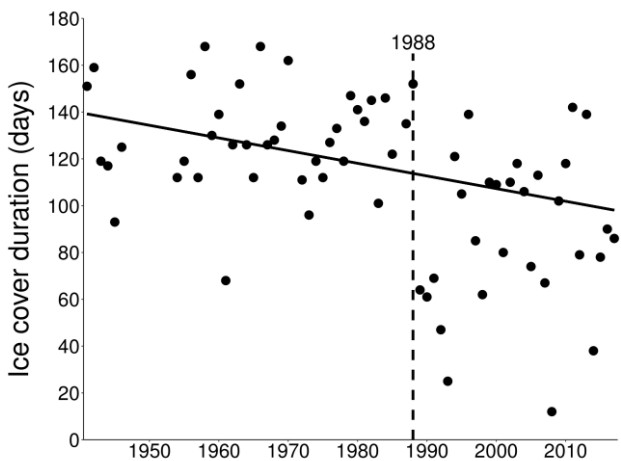

**Figure 11: Observed ice cover duration of Lake Erken between 1941-2017 (Sen's slope: -7.343 days/decade, p < 0.001). The dashed line shows the year (1988) of abrupt change in ice cover duration (Pettitt test, p < 0.001).**