# Peer review of "Historical modelling of changes in Lake Erken thermal conditions"

_Hydrology and Earth System Sciences, 2019_

## Referee Comment (RC1) · Chenxi Mi (Referee) · 3 Jun 2019

**General comments:**

This manuscript describes a one-dimensional model (i.e. GOTM) study for Lake Erken to analyze long-term changes of its thermal structure. As a major result, the model shows good performance in reproducing Lake Erken's past water temperatures and based on the statistical analysis, the lake temperatures rose rapidly during last fifty years, especially in the recent decades.

This is an interesting and important study, which could be considered for publication after a minor revision. Given the high simulation accuracy (based on the low RMSE), the model is capable in reconstructing lake temperatures for a long-term period, which provides a valuable database to analyze its changes under the climate warming. Based on my point of view, this paper is well organized and its content, especially the discussion part is valuable for the further development of GOTM. Detailed comments are shown below.

**Detailed comments:**

L10 in page 3: daily precipitation was used in driving the model, while the other six datasets were put into the model as hourly resolution. This sounds strange to me. Are different climate variables allowed to put into the model with different temporal resolution?

L30 in page 4: Why the measured water temperatures with 30 minutes resolution were averaged to daily, not the hourly mean values for the model calibration? In this way, the diurnal variation of the water temperature is missing. Could you give an explanation here?

L 3 in page 5: I am afraid the wind factor of 1.28 is a little bit high, since wind is measured in or quite close to the lake (based on Figure 1). Could you explain why you use such a high wind factor here?

L1 in page 6: how did you define the thermocline depth in the study? As I know, there are two ways in defining the thermocline depth in rLakeAnalyzer (i.e. seasonal=TRUE/FALSE). The results, from the two approaches, are different (see "Read, J. S., Hamilton, D. P., Jones, I. D., Muraoka, K., Winslow, L. A., Kroiss, R., Wu, C. H. & Gaiser, E. (2011). Derivation of lake mixing and stratification indices from high-resolution lake buoy data. Environmental Modelling and Software 26:1325-1336").

L4 in page 9: As stronger evidence for such changing trend, could you also use the measured water temperature to do a Mann-Kendall test? In the paper, all the statistical tests are based on the simulated

temperature, it is better to prove the simulated trend also based on the measured values. If it takes you so much time to do this work for all the three cases (i.e whole-lake, epilimnion and hyplimnion), I recommended to test the observed trend for the summer epilimnion, because the simulated temperatures of the layer significantly increased in the whole period.

L7 in page 12: I am confused here, you said that the summer epilimentic temperature significantly increased for the whole period, but not significantly increased in two sub-intervals? To me, it sounds like a paradox. Please check it.

Also, as shown in Blenckner 2002, Lake Erken is always ice-covered for the whole winter and the ice melts between March and early May. It is a weak point to use GOTM, without a ice module, to simulate such a lake with a long ice duration. I suggest adding some sentences, in this part, to clarify this limitation. Considering the future model development, it is a valuable work to include ice part into GOTM which could also be added into the Discussion.

Yours sincerely

Chenxi Mi at Magdeburg

---

## Referee Comment (RC2) · Roman Zurek (Referee) · 7 Jul 2019

The manuscript makes a significant contribution to the progress of science. The mathematical methods used showed the possibility of using even incomplete data.

Scientific quality: Are the scientific approach and applied methods valid? Are the results discussed in an appropriate and balanced way (consideration of related work, including appropriate references)? Scientific approach and applied methods valid for understanding climate change in relation to the functioning of stratified lakes. References are appropriate. All quotations in the text are given in References. The results are presented clearly and precisely. However, the reviewer lacks comparisons with other similar studies in the discussion like here and other similar: Report: Great Lakes

feeling effects of rapid climate warming (Update). https://phys.org/news/2019-03-great-lakes-effects-rapid-climate.html Steven Sadro John M. Melack James O. Sickman , Kevin Skeen 2018. Climate warming response of mountain lakes affected by variations in snow. https://doi.org/10.1002/lol2.10099 Vincent W.F. 2009. Effects of Climate Change on Lakes, Laval University, Quebec. W.M Mooij et al. 2005. The impact of climate change on lakes in the Netherlands: A review Aquatic Ecology 39(4):381-400 Skowron R. 2017.Water temperature in investigations of Polish lakes. Limnol. Rev. (2017) 17, 1: 31–46 It is valuable to discuss the biological effects of hypo- and epilimnion warming. Rating excellent.

The structure of the articles is clear, well-documented, language concise, it can sometimes be too concise. Drawings well document the theses of the article.

Increase letters in the X and Y axis descriptions

---

## Referee Comment (RC3) · Annie Visser (Referee) · 22 Jul 2019

| # | Page | Line | Comment |
|---|---|---|---|
| Major1 | 2 | 5-8 | This is simply not true. Stefan et al., 1998, include a section called model adequacy tests; Taner et al., 2011, state that they use a previously calibrated model (validation results not shown); and Winslow et al., 2017, include a section called technical validation.

Indeed it would be extremely bad practice to parameterise a model without calibration-validation. Please make your intentions and motivations for the study here much clearer.

This would also be improved by expanding the current introduction. At present there are only two paragraphs which cover very little literature. There is a need to root this work within the wider |
| Major2 | 4 | 1-22 | detailing the quantity and quality of the data would be extremely helpful.

Why are the additional sites only considered when there is missing data? Are the limited stations used truly representative of conditions across the entire lake? Would the coverage and overall consistency of the observed data not be improved if the same data was used at all times? Please clearly justify your decision-making here - deliberately excluding valid data is problematic.

Similarly, please indicate the locations of the additional stations on your map. It would be useful if the reader could understand the locations of these additional stations relative to the three detailed. |
| Major3 | 5 | 13-15; 21-24 | standard is to employ a split-sampling approach. Further, the aim is to minimise the variance in the GOF statistics across the calibration-validation period. Without defined periods, you cannot determine the consistency of the model performance.

L21 - What is meant by best? Was the algorithm run multiple times? How is the best one determined when three GOF statistics are used? Please clarify.

You introduce figures 2-4 but provide no further commentary on these. There is no clear discussion with regards to how this indicates good performance. Indeed, you do not refer to your GOF statistics through these figures at all. The reporting of the model performance needs to be significantly expanded. Please also consider reporting model performance per month and/or season - this may help to give insights into whether the model performs worse immediately following the ice-cover period.

Please also note that Figures 3 and 4 do not actually add anything to the reporting of model performance - they give no indication of the GOF of the model. Additionally, the use of inconsistent |
| Major4 | 5 | 26-28 | Please consider expanding on the limitation of ice-cover - perhaps in the discussion? For example, it would be helpful to suggest how this might be addressed, being unable to account for almost six months of the year is problematic. Similarly, this should be acknowledged int he section where you |
| Major5 | 9 | Figure 5 | x-axis to the start and end-year.

A continuous line should not be used to represent point data (single seasons per year). This data should be represented as points, or as a single continuous line containing all months.

Finally, please add space between the figures and their titles - at present it looks like the plot titles are related to the dashed line. Including the dashed and solid line in the legend would help. Three |
| Minor 9 | 2 | 9 | Define what is meant by a long record, for hydrological modelling of rivers this would mean > 30 |
| Minor1 | 1 | 6 | Need to explain what the abbreviation is - for example, consider: "General Ocean Turbulence model (GOTM), a hydrodynamic model configured in lake mode". |
| Minor10 | 2 | 10-11 | Please provide a citation if making a claim such as this. |
| Minor11 | 2 | 17 | What does significant mean? How much? Can you give a percentage or some other kind of |
| Minor12 | 2-3 | 29-30; 1;2 | Extremely limited detail on why the lake was considered - why should the reader care about the results from this particular work? What is interesting about it?

More information on the case study would also be useful. For example, an overview of the average climate, seasonality, the ecology of the area and anthropogenic influences. |

| Minor13 | 3 | 2 | What months represent winter? The reader cannot tell how many months the lake is actually ice-covered. Also, please clarify if it is the entirety of the lake which is ice-covered. |
|---|---|---|---|
| Minor14 | 3 | 9-12 | Repetitive - could simplify to say: "The model utilises six of these climatic parameters (excluding DP) |
| Minor15 | 3 | 4-13 | More information is required for GOTM. Why choose this model specifically? i.e. why is it well-suited for this application? Please also describe the structure of the model, what key processes does it capture? Define and describe the parameters of the model (Table 1). What are the limitations of the model? |
| Minor16 | 3 | Figure 1 | I am aware that the images used for review are not the final high-resolution images. However, this map looks equivalent to a screenshot. A north arrow and, critically, a scale bar, are missing. Additionally, labelling of features such as the roads and the island are unnecessary. Please consider producing a map using GIS Or similar software (mapping options are available in R). A map of |
| Minor17 | 3-4 | 15-17; 1-3 | Inconsistent use of meteorological station and weather station - please be consistent. For conciseness, the authors could simply state: "Driving climatic parameters were retrieved from |
| Minor18 | 3-4 | 15-17; 1-3 | Clearer signposting is required, please refer to the letters that each station represents in the main |
| Minor19 | 3 | 15 | Primarily retrieved from? What does primarily mean specifically? |
| Minor2 | 1 | 9 | Real is not very clear - consider replacing with "observed" (or similar). |
| Minor20 | 3 | 16 | Is the Malma weather station the Erken laboratory meteorological station? This inconsistency is |
| Minor21 | 4 | 10; 21 | What is meant by best? Please clarify how this is judged. |
| Minor22 | 4 | 24-25 | Is the lake always ice-free April-November? Additionally, please replace was with "is" - I presume that the ice-free period has not recently changed, therefore this should be in the present-tense. |
| Minor23 | 4 | 24-32 | Why is this text part of model calibration? This is still text relating to the input data. Perhaps consider combining 2.3 Data sources of driving parameters-2.3 Model calibration (paragraph 1) into a single Data section. |
| Minor24 | 5 | 1-2 | As with the hydrodynamic model, the reader needs to know why this approach is used. What is the |
| Minor25 | 5 | 5-7 | These lines are unclear, please consider rewording. |
| Minor26 | 5 | 8-13 | The authors state that an algorithm is used in the parameterisation. What is the stopping criteria? |
| Minor27 | 5-6 | 28; 1-3 | Please explain to the reader why they should care about these metrics - why are they important? |
| Minor28 | 6 | 12 | Did you test for autocorrelation? Was it all autocorrelated? Please be clearer. |
| Minor29 | 6 | 13-14 | Please correct Figure 3 accordingly - the time-series should not extend beyond the point for which it |
| Minor3 | 1 | 10-12 | Suggest the author's state why the results are split into these sub-intervals; until very late on the paper I presume the split was because pre-1988 records were patchy. |
| Minor30 | 6 | 21 | Please consider moving this line to the start of the section. Please also include the package version |
| Minor31 | 9 | 2 | What data did this use? The pre-1988 data which included data from mixed stations and the post-1988 data which was much more consistent? Can this finding be trusted? |
| Minor32 | 9 | 2 | In the discussion, please explain to the reader why this matters, what it indicates etc. - It is not |
| Minor33 | 9 | 5 | Please define your terms, e.g. epilmnetic. |
| Minor34 | 9 | 9-14 | Please be consistent in the number of significant figures for temperature. |
| Minor35 | 9 | 14 | Please start a new paragraph before discussing thermal stratification. |
| Minor36 | 9-10 | | As a decadal mean, it would be useful to see the reporting of confidence intervals for these values. |
| Minor37 | 11 | 1-8 | You cannot claim that there was a good match. No valid assessment of model performance was provided. This needs to be significantly addressed before such a claim can be asserted. |
| Minor38 | 11 | 12 | I do not agree that it indicates the reliabilty, the wording is too strong. It could be described as a |
| Minor39 | 12 | 1-11 | Much of this text appears to be results. |
| Minor4 | 1 | 10/11/15/16 | State the months associated with your seasons. |
| Minor40 | 12 | 14-17 | The provision of a confidence interval would help to expand upon this further (it could also improve |
| Minor41 | 12 | 18-22 | Does O'Reilly account for the influence of ice-cover? If yes, could this not also account for some of the discrepancy? Please weight the pros and cons of this study versus theirs accordingly. |
| Minor42 | 13 | 25-34 | Suggest that the authors consider leading the discussion with this text. At present, it is not clear to the reader why this work or the results is relevant - the implications are not made clear. |
| Minor43 | 14 | 8-12 | The assertion of "accurately" cannot be made whilst there is no robust consideration of model |
| Minor44 | 14 | 13-19 | Suggest that a dedicated conclusion would help to wrap up the paper and reassert the |
| Minor5 | 1 | 23-25 | Abstract does not necessarily make clear why this matters - what is the need for the work? |
| Minor6 | 1 | 27-29 | This first sentence is repetitive; also not convinced that Samal et al., 2012 is the best citation for this critical statement - there are other more relevant seminal works that the authors may cite. |

| | | | |
|---|---|---|---|
| Minor7 | 2 | 1-2 | Again, repetition - it is self-evident that a rise in lake water tempeature increases water |
| Minor8 | 2 | 5-7 | It would be helpful to explain what some of the conclusions of these studies are/were - it makes it |
| NA | 6 | 4-7 | I would like to highlight that the level of description here is excellent and represents the level that |

---

## Author Comment (AC2) · 26 Aug 2019

We would like to thank Roman Zurek (Referee) for the comments he provided on our manuscript. We will take into consideration expanding our Discussion section following your suggestion. Moreover, the quality of the figures will be improved.

On behalf of all the authors, Simone Moras

---

## Author Comment (AC3)

Authors' response to Referee 3

**Referee (1) – Page 2, lines 5-8.** I do not understand the claim that these studies do not use observations to validate their models. This is simply not true. Stefan et al., 1998, include a section called model adequacy tests; Taner et al., 2011, state that they use a previously calibrated model (validation results not shown); and Winslow et al., 2017, include a section called technical validation. Indeed it would be extremely bad practice to parameterize a model without calibration-validation. Please make your intentions and motivations for the study here much clearer.
This would also be improved by expanding the current introduction. At present there are only two paragraphs which cover very little literature. There is a need to root this work within the wider research.

**Authors' response (1) – Page 2, lines 5-8.** This is bad wording on our part. We did not mean to imply that the mentioned studies did not validate their models. What we were trying to say was that these studies have focused on simulating future changes in lake thermal structure (with model validation to present conditions) while in this paper we are advocating for running simulations farther back in time than is normally done for validation purposes in order to provide evidence that climate change has already affected lake thermal structure.
We propose to expand our introduction by referencing to papers that have investigated historical record changes in lake water temperature (e.g. Kainz et al. 2017, Livingstone et al. 2001).

**Referee (2) – Page 4, lines 1-22.** How much data is actually missing from the dataset for each parameter? A table, or similar, detailing the quantity and quality of the data would be extremely helpful.
Why are the additional sites only considered when there is missing data? Are the limited stations used truly representative of conditions across the entire lake? Would the coverage and overall consistency of the observed data not be improved if the same data was used at all times? Please clearly justify your decision-making here - deliberately excluding valid data is problematic.
Similarly, please indicate the locations of the additional stations on your map. It would be useful if the reader could understand the locations of these additional stations relative to the three detailed.

**Authors' response (2) – Page 4, lines 1-22.** A detailed description of the number of missing data is available in the supplementary material (tables 1-4). We put these tables in the supplementary material for a better readability of the paper. However, we are considering to add a single summary table in the paper.
Our meteorological data is either collected from a small island (A fig1) in the lake or from a meteorological station only a few hundred meters from the lake shore (B fig 1). Given the station locations, we considered this ideal data for forcing a lake model and it was our assumption that these data should be used when available. When data were missing, we found that the neural network models made use of as many of the surrounding data sources as possible providing the most accurate replacement values. We do not believe that we were excluding valid data, based on our belief (and we suspect a widely held belief) that locally collected data would be most appropriate for modeling. Data from additional sites was only used as a substitute when the most valid data were not available.
We are also considering adding a map with the meteorological station we used in our study.

**Referee (3) – Page 5, lines 13-15; 21-24.** The authors appear to use the same data for calibration-validation. Why is this? Please justify - the standard is to employ a split-sampling approach. Further, the aim is to minimize the variance in the GOF statistics across the calibration-validation period. Without defined periods, you cannot determine the consistency of the model performance.
L21 - What is meant by best? Was the algorithm run multiple times? How is the best one determined when three GOF statistics are used? Please clarify.
You introduce figures 2-4 but provide no further commentary on these. There is no clear discussion with regards to how this indicates good performance. Indeed, you do not refer to your GOF statistics through these figures at all. The reporting of the model performance needs to be significantly expanded. Please also consider reporting model performance per month and/or season - this may help to give insights into whether the model performs worse immediately following the ice-cover period.
Please also note that Figures 3 and 4 do not actually add anything to the reporting of model performance - they give no indication of the GOF of the model. Additionally, the use of inconsistent x-y scales is bad practice and misleading. If producing the figures in R then it is possible to fix the axes across plots/facets.
As a more minor comment - it is not necessary to define the three equations, tehy are standard mathematical equations. What is more important is to explain why these are relevant - what insight does using these GOF statistics provide?

**Authors' response (3) – Page 5, lines 13-15; 21-24.** We agree that for typical applications of models where the goal is to make simulations to future or otherwise different conditions than are covered by the record of measured calibration data it is appropriate to employ a split calibration and validation strategy. However in our case the goal was not to simulate outside of the period of available calibration data, but to use the model to provide a complete and consistent record over a period in which calibration data were available but incomplete (especially in the earlier part of the record). In such a case we believe it is better to make full use of all measured calibration data rather than removing some for a separate validation run. This should ensure that the calibration encompasses the widest possible range of variability and provides parameter values that are most appropriate for the entire period simulated in our study.
L21 - In the ACPy calibration the best set of parameter is calculated by minimizing the log likelihood function.
The caption of figures 2-4 will be expanded and there will be a reference to the model performance which are currently described in table 1 (page 18). The use of inconsistent x-y scales on fig. 3-4 will be fixed in the revised paper. We propose two solutions: the first one is to make the x-axis of the same lenght in each panel (the same number of years will be shown). Alternatively, we could show the comparison between observed and modelled water temperature of some representative years.
The standard mathematical equations will be removed in the revised paper.

**Referee (4) – Page 5, lines 26-28.** Please consider expanding on the limitation of ice-cover - perhaps in the discussion? For example, it would be helpful to suggest how this might be addressed, being unable to account for almost six months of the year is problematic. Similarly, this should be acknowledged in the section where you describe GOTM. For instance, why do you not simply use a model that does include an ice-cover module given the length of time the lake is ice-covered?

**Authors' response (4) – Page 5, lines 26-28.** The GOTM model used for the simulations documented here did not have a functioning ice model, but instead cut off surface heat exchange when the

simulated surface water temperature became negative.  This provided a very simple way to make continuous simulations that include freezing conditions that would normally lead to the formation of ice.  However, the temperature profiles during winter were not realistic, and could not be used for model calibration.   This can be seen in figures 1-2 (below) where a comparison between simulated an observed water temperature at 1m and 15 m depth is reported for year 2009. At 1 m depth, simulated and observed temperature are rather similar throughout the entire year. However, at 15 m depth, the model does not take into account the heat loss from sediment during ice-cover, which cause an increase in bottom water temperature. During winter, there is a clear mismatch between simulated and observed water temperature. For this reason, all data collected between 1 December - 31 March are excluded from the temperature data used for model calibration and only data between 1 April - 30 November are used for calibrating the model.  .

Moreover, the reason why we used GOTM model is that this paper is the result of a spin-off study within the PROGNOS project (http://prognoswater.org/) which uses this model to provide real-time predictions of water quality using short-term weather forecast data. In this study, the GOTM model was tested for Lake Erken which is part of PROGNOS. This model has the advantage that it can be coupled to biogeochemical models, which is crucial for the aims of PROGNOS.

We will follow your valuable suggestion to expand the limitation of the ice-cover in our discussion.

[Figure]

**Figure 1**

[Figure]

**Figure 2**

**Referee (5) – Page 9, figure 5.** Again, please use consistent y-axes and begin at zero. This is bad practice and misleading. Limit the x-axis to the start and end-year.
A continuous line should not be used to represent point data (single seasons per year). This data should be represented as points, or as a single continuous line containing all months.
Finally, please add space between the figures and their titles - at present it looks like the plot titles are related to the dashed line. Including the dashed and solid line in the legend would help. Three duplicated legends are not necessary, replace with a single legend.

**Authors' response (5) – Page 9, figure 5.** We will improve the readability and consistency of the plots following your suggestions.

**Referee (6) – Page 2, line 9.** Define what is meant by a long record, for hydrological modelling of rivers this would mean > 30 years, for hydroecology > 15 years is considered long.
**Authors' response (6) – Page 2, line 9.** From our point of view the definition of long record is somewhat arbitrary. We think that a long record should encompass the historical changes in climate that have already occurred. For example, a record of 50 years of data can be considered long.

**Referee (7) – Page 1, line 6.** Need to explain what the abbreviation is - for example, consider: "General Ocean Turbulence model (GOTM), a hydrodynamic model configured in Lake Mode".

**Authors' response (7) – Page 1, line 6.** Thank you for the suggestion. This will be added in the abstract.

**Referee (8) – Page 2, lines 10-11.** Please provide a citation if making a claim such as this.

**Authors' response (8) – Page 2, lines 10-11.** This is based on the experience of one author (Don Piesrond) who has worked for public water utilities and found that effects of climate change that have already occurred support policies that mitigate future expected changes. To our knowledge, no reference is available.

**Referee (9) – Page 2, line 17.** What does significant mean? How much? Can you give a percentage or some other kind of numerical indication?

**Authors' response (9) – Page 2, line 17.** The word "significant" will be changed to "large". Moreover, we are considering adding a table with the number of available daily water temperature data for Lake Erken and number of missing data in the interval 1961-2017.

**Referee (10) – Pages 2-3, lines 29-30; 1-2.** Extremely limited detail on why the lake was considered. Why should the reader care about the results from this particular work? What is interesting about it? More information on the case study would also be useful. For example, an overview of the average climate, seasonality, the ecology of the area and anthropogenic influences.

**Authors' response (10) – Pages 2-3, lines 29-30; 1-2.** Lake Erken has been extensively studied in the last 70 years and it has a considerable amount of water temperature data available, which made it a good study case for testing the methodological approach of this paper. Moreover, we think that this paper is addressed to describe a valid methodology that aim to reconstruct past water temperature of lakes and not to the ecological importance of Lake Erken in itself.
We will consider providing more information on the investigated lake during the paper revision.

**Referee (11) – Page 3, line 2.** What months represent winter? The reader cannot tell how many months the lake is actually ice-covered. Also, please clarify if it is the entirety of the lake which is ice-covered.

**Authors' response (11) – Page 3, line 2.** In this paper, we considered the period of Dec-Mar as winter, when the lake is ice-covered. However, in some years, the onset of ice-cover starts in January and it ends in April. Yes, the lake is ice-covered in its entirety. This information will be added in the revised paper.

**Referee (12) – Page 3, lines 9-12.** Repetitive - could simplify to say: "The model utilises six of these climatic parameters (excluding DP) at an hourly timestep; DP is input on a daily timestep."

**Authors' response (12) – Page 3, lines 9-12**. Thank you for the suggestion. This will be modified in the revised paper.

**Referee (13) – Page 3, lines 4-13.** More information is required for GOTM. Why choose this model specifically? Why is it well-suited for this application? Please also describe the structure of the model,

what key processes does it capture? Define and describe the parameters of the model (Table 1). What are the limitations of the model?

It is also worth stating that GOTM, and all the other software/codes used, are Open Source.

**Authors' response (13) – Page 3, lines 4-13.** GOTM is mainly used as a stand-alone model for hydrodynamic applications in natural water, such as surface heat fluxes, surface mixed-layer dynamics and stratification processes.

The adjusted model parameters in this study are heat-flux, shortwave radiation and wind factors which are non-dimensianal scaling factor that are adjusted to minimize the difference between observed and modelled temperature. The minimum turbulent kinetic energy (k_min) and the e-folding depth for visible fraction (g2) are parameters that strongly influence the vertical distribution of light and temperature in the water column. Low values of g2 represent a higher exinction coefficient promoting higher surface temperature.

A known limitation of the model is the lack of an ice-module and a complete energy balance of the ice including ice growth and ice decay is not calculated by GOTM at this time.

We will state that GOTM is an OpenSource model in the revised paper.

**Referee (14) – Page 3, figure 1.** I am aware that the images used for review are not the final high-resolution images. However, this map looks equivalent to a screenshot. A north arrow and, critically, a scale bar, are missing. Additionally, labelling of features such as the roads and the island are unnecessary. Please consider producing a map using GIS Or similar software (mapping options are available in R). A map of Sweden indicating the location of the lake, which would highlight the relative scale, are also necessary.

**Authors' response (14) – Page 3, figure 1.** This is a very good point. Yes, we are aware that the map in the discussion paper is not ideal. We will improve this map in the revised paper.

**Referee (15) – Pages 3-4, lines 15-17; 1-3.** Inconsistent use of meteorological station and weather station - please be consistent. For conciseness, the authors could simply state: "Driving climatic parameters were retrieved from meteorological stations at…".

**Authors' response (15) – Pages 3-4, lines 15-17; 1-3.** Thanks for the suggestion. The naming will be made consistent in the revised paper.

**Referee (16) – Pages 3-4, lines 15-17; 1-3.** Clearer signposting is required, please refer to the letters that each station represents in the main body text.

**Authors' response (16) – Pages 3-4, lines 15-17; 1-3.** This will be corrected in the revised paper.

**Referee (17) – Page 3, line 15.** Primarily retrieved from? What does primarily mean specifically?

**Authors' response (17) – Page 3, line 15.** We propose to modify the sentence as follows: "Driving meteorological parameter were retrieved whenever possible from the Erken laboratory meteorological station…"

**Referee (18) – Page 1, line 9.** Real is not very clear - consider replacing with "observed" (or similar).

**Authors' response (18) – Page 1, line 9.** Thanks for the suggestion. This will be corrected in the revised paper.

**Referee (19) – Page 3, line 16.** Is the Malma weather station the Erken laboratory meteorological station? This inconsistency is reflected in the caption as well.

**Authors' response (19) – Page 3, line 16.** Yes, it is. This will be better clarified in the revised paper.

**Referee (20) – Page 4, lines 10; 21.** What is meant by best? Please clarify how this is judged.

**Authors' response (20) – Page 4, lines 10; 21.** We propose to remove lines 8-14 (page 4) and re-write them as follows: "To made maximum use of data from surrounding station we used Artificial Neural Network function fitting analysis (ANN nftool) to predict missing meteorological data at Erken. The analysis was carried out using MATLAB version R2017b (MathWorks Inc. Natick, Massachussets)". We propose to modify lines 20-21 (page 4) into: "Offsite and local dataset overlap for at leat 8-10 years to get a reasonable number of data to perform ANN function fitting analysis that describes input-target relationship."

**Referee (21) – Page 4, lines 24-25.** Is the lake always ice-free April-November? Additionally, please replace was with "is" - I presume that the ice-free period has not recently changed, therefore this should be in the present-tense.

**Authors' response (21) – Page 4, lines 24-25.** This period is usually longer than the total period of ice cover which can be variable from year to year.  There are occasions when ice continues into end of April, but the mid-April to November period is definitely representative of ice free conditions and using data from this period to calibrate the model will definitely avoid errors associated with GOTM's simplistic simulation of ice cover. During the revision of the manuscript we noticed we noticed that water temperature data of April and November were not considered to calibrate the model. We re-run the model including April and November water temperature and we updated the calibration parameters. The new calibration provided very similar results to the calibration showed in the discussion paper. These are the values of the updated calibrated parameters and model statistics
- Heat-flux factor: 0.863009
- Short- wave radiation factor: 0.970753
- Wind factor: 1.28701
- Minimum turbulent kinetic energy: 1.64873e-06
- e-folding depth for visible fraction: 2.63732

- ln Likelihood: -60469.715
- Bias (°C): -0.04707
- MAE (°C): 0.7529
- RMSE (°C): 1.089
- Correlation: 0.9717

Table 1 will be updated with these new values in the revised paper.
The sentence will be corrected with the present- tense.

**Referee (22) – Page 4, lines 24-32.** Why is this text part of model calibration? This is still text relating to the input data. Perhaps consider combining 2.3 Data sources of driving parameters-2.3 Model calibration (paragraph 1) into a single Data section.
Additionally, there is an issue with the section numbering: 2.3-2.4-2.3

**Authors' response (22) – Page 4, lines 24-32.** We are considering to rename the section "2.3 Data sources of driving parametes"(page 3, line 14) into "2.3 Data sources of meteorological parameters" and rename the section "2.3 Model calibration" (page 4, line 23) into "2.5 Observed water temperature data and model calibration".
We will correct the section numbering in the revised paper.

**Referee (23) – Page 5, lines 1-2.** As with the hydrodynamic model, the reader needs to know why this approach is used. What is the rationale?

**Authors' response (23) – Page 5, lines 1-2.** We propose to add the following sentence in the text: **"**ACPy is a utility that eliminates the need for time consuming manual calibration of hydrodynamic and water quality models.  This allows for more extensive testing and evaluation of model calibrations, ultimately providing more accurate and repeatable results".

**Referee (24) – Page 5, lines 5-7.** These lines are unclear, please consider rewording.

**Authors' response (24) – Page 5, lines 5-7.** We propose to modify lines 5-7 as follows: "Simulations were run between 1961 and 2017 but in order to obtain stable initial conditions the model was run for an additional one year spin up using a copy of the 1961 data. In this way, 1961 data were both used as spin-up year and discarded from calibration and then reused in the proceeding calibration."

**Referee (25) – Page 5, lines 8-13.** The authors state that an algorithm is used in the parameterisation. What is the stopping criteria? How does the algorithm select a parameter set?

**Authors' response (25) – Page 5, lines 8-13.** ACPy was set to run 10000 simulation during calibration to get a stable solution to obtain the optimal parameter set that minimizes the log likelihood function.

**Referee (26) – Pages 5-6, lines 28; 1-3.** Please explain to the reader why they should care about these metrics - why are they important? What do they indicate?

**Authors' response (26) – Pages 5-6, lines 28; 1-3.** We propose to rephrase lines 26-27 (page5) as follows: "We summarized the model temperature output by calculating a number of statistics that can qualify the ecological consequence of changes in thermal stratification using Lake Analyzer R

Package. The ecological implications of the changes of these metrics due to climate change are discussed in detail in the Discussion section.

**Referee (27) – Page 6, line 12.** Did you test for autocorrelation? Was it all autocorrelated? Please be clearer.

**Authors' response (27) – Page 6, line 12.** Yes, we tested for autocorrelation using acf and pacf function in R. The modified Mann-Kendall test was used for detecting monotonic increase of stratification length and termination and growing season.

**Referee (28) – Page 6, lines 13-14.** Please correct Figure 3 accordingly - the time-series should not extend beyond the point for which it is useful!

**Authors' response (28) – Page 6, lines 13-14.** Thanks for the suggestion. This can be improved in the revised paper.

**Referee (29) – Page 1, lines 10-12.** Suggest the author's state why the results are split into these sub-intervals; until very late on the paper I presume the split was because pre-1988 records were patchy.

**Authors' response (29) – Page 1, lines 10-12.** Thanks for the suggestion. We will add a sentence in the abstract that specifies that the splitting was selected because of an abrupt change in air temperature.

**Referee (30) – Page 6, line 21.** Please consider moving this line to the start of the section. Please also include the package version for Lake Analyzer (the citation seems relatively old).

**Authors' response  (30) – Page 6, line 21.** The line will be moved at the beginning of the section. The package version used here is 1.11.4 and the citation will be modified as with the following: "Winslow, L., Read, J., Woolway, R., Brentrup, J., Leach, T.,Zwart, J., Albers, S., and Collinge, D: rLakeAnalyzer: Lake Physics Tools. R package version 1.11.4. https://CRAN.R-project.org/package=rLakeAnalyzer, 2018."

**Referee (31) – Page 9, line 2.** What data did this use? The pre-1988 data which included data from mixed stations and the post-1988 data which was much more consistent? Can this finding be trusted?

**Authors' response (31) – Page 9, line 2.** Yes, pre-1988 data mostly included data from mixed station. However, these data have been adjusted to better represent Lake Erken local conditions (at Malma met station) using Neural Network function fitting analysis. This analysis report a very good agreement between Erken air temperature (target data) and output data (see suplementary material). Given that, we think that we think that air temperature dataset used in this study is reliable.

**Referee (32) – Page 9, line 2.** In the discussion, please explain to the reader why this matters, what it indicates etc. - It is not made clear.

**Authors' response (32) – Page 9, line 2.** From our point of view, an abrupt change in air temperature support the fact that a more rapid change in water temperature is occurring in the last decades and that the effect of climate change on thermal conditions is accelerating.

**Referee (33) – Page 9, line 5.** Please define your terms, e.g. epilmnetic.

**Authors' response (33) – Page 9, line 5.** We will consider to define the terms in the revised paper.

**Referee (34) – Page 9, lines 9-14.** Please be consistent in the number of significant figures for temperature.

**Authors' response (34) – Page 9, lines 9-14.** Thanks for the suggestion. This will be corrected in the revised paper.

**Referee (35) – Page 9, line 14.** Please start a new paragraph before discussing thermal stratification.

**Authors' response (35) – Page 9, line 14.** Thanks for the suggestion. This will be corrected in the revised paper.

**Referee (36) – Pages 9-10.** As a decadal mean, it would be useful to see the reporting of confidence intervals for these values. Perhaps consider a table of results.

**Authors' response (36) – Pages 9-10.** Thanks for your suggestion. We will consider to add a table of result in the revised paper.

**Referee (37) – Page 11, lines 1-8.** You cannot claim that there was a good match. No valid assessment of model performance was provided. This needs to be significantly addressed before such a claim can be asserted.

**Authors' response (37) – Page 11, lines 1-8.** In the revised paper we will refer to the table and figures where the model performance is reported.

**Referee (38) – Page 11, line 12.** I do not agree that it indicates the reliabilty, the wording is too strong. It could be described as a positive indication.

**Authors' response (38) – Page 11, line 12.** We propose to change the word "reliability" into "consistency".

**Referee (39) – Page 12, lines 1-11.** Much of this text appears to be results.

**Authors' response (39) – Page 12, lines 1-11.** From our point of view, reporting again the major results in the discussion section and provide a possible explanation for them would be easier to follow/understand for the reader and would improve the readability of the paper.

**Referee (40) – Page 1, lines 10-11; 15-16.** State the months associated with your seasons.

**Authors' response (40) – Page 1, lines 10-11; 15-16.** Thanks for your suggestion. The months will be added in the revised paper.

**Referee (41) – Page 12, lines 14-17.** The provision of a confidence interval would help to expand upon this further (it could also improve or worsen the difference in results).

**Authors' response (41) – Page 12, lines 14-17.** A confidence interval is not reported by O'Reilly et al. (2015), but only the Sen's slope of lake summer water temperature trends.

**Referee (42) – Page 12, lines 18-22.** Does O'Reilly account for the influence of ice-cover? If yes, could this not also account for some of the discrepancy? Please weight the pros and cons of this study versus theirs accordingly.

**Authors' response (42) – Page 12, lines 18-22.** Trends reported by O'Reilly et al. do not account for ice-cover since the work reports lake surface summer temperature trends. However, the paper does state that lakes that are always completely ice-covered during winter (this is the case of Lake Erken) are experiencing a faster warming trend compared to lakes that do not freeze during winter.

**Referee (43) – Page 13, lines 25-34.** Suggest that the authors consider leading the discussion with this text. At present, it is not clear to the reader why this work or the results is relevant - the implications are not made clear.

**Authors' response (43) – Page 13, lines 25-34**. We consider that the first part of the discussion is useful to demonstrate the validity of the first aim of this work that is to provide a valuable and reliable method to extend historical water temperature records back in time. In the second part (lines 25-34 page 13 and lines 1-7 page 14) we provided the most important ecological implications that a warmer climate might have on lake Erken specifically. This order follows the same order of how the aims are described in the introduction.

**Referee (44) – Page 14, lines 8-12.** The assertion of "accurately" cannot be made whilst there is no robust consideration of model performance.

**Authors' response (44) – Page 14, lines 8-12**. We propose to rephrase lines 8-9 (page 14) in the following way: "The present study has shown that the use of the GOTM model to reconstruct the past 57-years of thermal condition of Lake Erken was a valuable too for detecting changes in its thermal structure. This methodology presented here can be extended to other lakes that have a

record of water temperature data. The use of local meteorological data can be used to model water temperature record further back in time or fill data gaps in water temperature records."

**Referee (45) – Page 14, lines 13-19.** Suggest that a dedicated conclusion would help to wrap up the paper and reassert the aims/objectives and relevance of the work.

**Authors' response (45) – Page 14, lines 13-19.** Thanks for the suggestion. These lines will be moved into a dedicated conclusion in the revised paper.

**Referee (46) – Page 1, lines 23-25.** Abstract does not necessarily make clear why this matters - what is the need for the work?

**Authors' response (46) – Page 1, lines 23-25.** We propose to remove line 5 (page 1) and write as follows: "Historical lake water temperature records are a valuable source of information to assess the influence of climate change on lake thermal structure. However, in most cases such records span a short period of time and/or are incomplete, providing a less credible assessement of changes in lake water temperature."

**Referee (47) – Page 1, lines 27-29.** This first sentence is repetitive; also not convinced that Samal et al., 2012 is the best citation for this critical statement. There are other more relevant seminal works that the authors may cite.

**Authors' response (47) – Page 1, lines 27-29.** We propose to re-phrase lines 27-29 as follows: "Changes in the thermal structure and mixing regimes of lakes are a consequence of changes in several climatic factors such as air temperature, solar radiation, cloud cover, wind speed and humidity (Woolway et al.2019)."

**Referee (48) – Page 2, lines 1-2.** Again, repetition - it is self-evident that a rise in lake water temperature increases water temperature - please be more concise.
**Authors' response (48) – Page 2, lines 1-2.** Thanks for your suggestion. The citation of Arhonditsis et al. can be removed for conciseness.

**Referee (49) – Page 2, lines 5-7.** It would be helpful to explain what some of the conclusions of these studies are/were - it makes it clearer to the reader why there is a need for this.

**Authors' response (49) – Page 2, lines 5-7.** Thanks for your suggestion. We will consider to expand this part of introduction in the revised paper.

**Referee (50) – Page 6, lines 4-7.** I would like to highlight that the level of description here is excellent and represents the level that should be achieved throughout the manuscript.

**Authors' response (50) – Page 6, lines 4-7.** Thanks for your comment!

---

## Author Response (AR1)

**Revision of the manuscript "Historical modelling of changes in Lake Erken thermal conditions" by Moras S, Ayala A.I. and Pierson D.C.**

**Changes after manuscript revision**

5 We revised the manuscript following referees' suggestions. The major changes in the manuscript are the following:

- After noticing a small error in the model calibration in the first manuscript version (observed water temperature data of April and November were not taken into account during calibration) we updated all the tables and figures with the new results. This update, however does not change the main outcomes of the study.
- The introduction has been rewritten by adding references to studies that analyzed with historical lake water
10 temperature data. Moreover, we better addressed the aim of the study.
- We better clarified the model limitation in simulating ice cover in the method section.
- We added the analysis of observed ice cover duration of Lake Erken between 1941-2017. This analysis was not present in the first version of the manuscript.
- We reported the results of the model performance in different season to evaluate if the model performs differently
15 in different seasons.
- We added a paragraph discussing the model performance in the discussion section
- Three more tables have been added to the manuscript. One table shows how much data we retrieved from the different meteorological station for each meteorological parameter we used to drive the model. Another table shows the model performance of the model on seasonal basis. The third table we added is a table of results of the lake
20 metrics we investigated.
- All the figures have been update with the new results and we followed referees' suggestion to improve the readability of the figures.
- We added a new section in the supplementary material that describe the mismatch between modelled and observed water temperature in winter

For the detailed changes in the manuscript, please the the marked up version of the manuscript below.

**Authors' response to Referee 1**

We would like to thank Referee 1 for the valuable comments he provided for our manuscript, that contribute to
30 improve the quality of our work. See below detailed answers to the comments.

**Referee 1 – Page 3, line 10**. Daily precipitation was used in driving the model, while the other six datasets were put into the model as hourly resolution. This sounds strange to me. Are different climate variables allowed to put into the model with different temporal resolution?

**Authors' response**. The seven climatic parameters used to drive the models are grouped into three input datasets: a meteo_file,
35 which contains wind speed, air pressure, air temperature, cloud cover and relative humidity data; a swr_file in which shortwave radiation data are stored and a precip_file where precipitation data are located. Within the same dataset, the parameters must have the same time-resolution, but it is possible each of the three datasets to differ in time-resolution. GOTM model allows to set a factor that converts the unit of measurement used in the the precip_file input (in our case mm/day) into the unit of measurement used in GOTM for precipitation (m/s). This possibility gave us the chance to use the most suitable time resolution
40 for precipitation in our study, since no weather station around Lake Erken measured precipitation on hourly basis. For our long-term simulations we presented in our paper, we assume a constant water level. Therefore, precipitation had only minor effects on the model output.

**Referee 1 – Page 4, line 30**. Why the measured water temperatures with 30 minutes resolution were averaged to daily, not the hourly mean values for the model calibration? In this way, the diurnal variation of the water temperature is missing. Could you give an explanation here?

Authors' response. This is a good point and we are aware that using hourly values for model calibration would have taken into account the diurnal variation of water temperature. Our choice to average 30 minutes water temperature to daily values have been made by the fact that a calibration using hourly values was computationally too intensive. We set ACPy to run 10000 simulations to obtain the best parameter set. We calibrated the model using a daily water temperature dataset of 94244 data points. This process takes ~24 hours using daily values. The use of hourly data for model calibration would have been a very time-consuming process. In addition, most of the metrics of change in thermal structure used in our paper were most conveniently calculated using mean daily data. Therefore, we felt that it would be most appropriate to develop model calibration based on mean daily output.

**Referee 1 – Page 5, line 3**. I am afraid the wind factor of 1.28 is a little bit high, since wind is measured in or quite close to the lake (based on Figure 1). Could you explain why you use such a high wind factor here?

Authors' response. There are two possible explanations here. First, the dominant wind speed (ws) direction is along the longest east-west fetch of Lake Erken that is ~10 km as opposed to the north – south fetch that is only 2-3 km. The 1D model input for wind is only a mean velocity and does not account for the effects of fetch. Given that wind is often blowing along the longest fetch that would have that would have the greatest effect on the measured temperature measurements used for calibration at the Eastern end of the lake, it is reasonable to expect an elevated wind factor. Secondly, it is actually the wind speed cubed that is used in the model equations that effect turbulent mixing. Under variable and gusty conditions cubing the mean hourly wind speed calculated by our data logger measuring at 1 minute intervals

$$\left(\frac{\sum_{60}^{1} ws}{60}\right)^3$$

may underestimate the true effects of wind which would more properly be calculated as the mean of of all cubed wind speed measurements made during the hour.

$$\left(\frac{\sum_{60}^{1} ws^3}{60}\right)$$

This effect would also result in an elevated wind factor.

**Referee 1 – Page 6, line 1**. how did you define the thermocline depth in the study? As I know, there are two ways in defining the thermocline depth in rLakeAnalyzer (i.e. seasonal=TRUE/FALSE). The results, from the two approaches, are different (see " Read, J. S., Hamilton, D. P. P., Jones, I. D., Muraoka, K., Winslow, L. A., Kroiss, R., Wu, C. H. & Gaiser, E. (2011). Derivation of lake mixing and stratification indices from high resolution lake buoy data. Environmental Model ling and Software 26:1325 1336 ")").

**Authors' response**. We did not specify which condition I used to define thermocline depth in our R code. However, not specifying any condition as we did gives the same result of the condition "seasonal = TRUE".

**Referee 1 – Page 9, line 4**. As stronger evidence for such changing trend, could you also use the measured water temperature to do a Mann Kendall test? In the paper, all the statistical test s are based on the simulated temperature, it is better to prove the simulated trend also based on the temperature, it is better to prove the simulated trend also based on the measured values. If it takes you so much time to do this work for all the three cases (i.e whole lake, epilimnion and hyplimnion), I recommended to test the observed trend for the summer epilimnion because the simulated temperatures of the layer significantly increased in the whole period.

**Authors' response**. Even though Lake Erken has a relatively long measured water temperature record compared to other lakes, there are still significant data gaps within the dataset. There were several years with no (or very few) measured

temperature before the deployment of the automatic floating station in 1988. There are significant data gaps in Erken temperature record after 1988 as well, during the maintenance/failures of the floating station for example. Since our trend analysis is based on seasonal means, performing a trend analysis on measured water temperature with several missing data would have made our results unrealistic. Having such data gaps in our water temperature record is actually the main reason

5 why we developed the approach described in this study in order to get a more consistent and reliable water temperature historical record using a hydrodynamic model.

**Referee 1 – Page 12, line 7**. I am confused here, you said that the summer epilimentic temperature significantly increased for the whole period, but not significantly increased in two sub intervals? To me, it sounds like a paradox. Please check it.

**Authors' response**. When Mann-Kendall test is performed on the two sub-intervals (1961-1988 and 1989-2017) of summer

10 epilimnetic temperature, positive trends are detected but they are not significant. This means that the two sub-intervals are too short to detect a significant trend. Indeed, when the trend test is performed on the entire study period (1961-2017) the summer epilimnetic temperature shows a significant increasing trend. From our results, we can infer that summer epilimnetic temperature was subjected to a slower but more stable warming compared to, for example, spring and autumn epilimnetic temperature, which showed a more abrupt increase in water temperature in the most recent sub-interval (1989-2017).

**Referee 1**. Also, as shown in Blenckner 2002, Lake Erken is always ice covered for the whole winter and the ice melts between March and early May. It is a weak point to use GOTM, without an ice module, to simulate such a lake with a long ice duration. I suggest adding some sentences, in this part, to clarify this limitation. Considering the future model development, it is a valuable work to include ice part into GOTM which could also be added into the Discussion.

20 **Authors' response**. GOTM developers are currently working on integrating GOTM with an ice module, but this was not available for this work. The GOTM model used for the simulations documented here did not have a functioning ice model, but instead cut off surface heat exchange when the simulated surface water temperature became negative. This provided a very simple way to make continuous simulations that include freezing conditions that would normally lead to the formation of ice. However, the temperature profiles during winter were not realistic, and could not be used for model calibration. This can

25 be seen in figures 1-2 (below) where a comparison between simulated an observed water temperature at 1m and 15 m depth is reported for year 2009. At 1 m depth, simulated and observed temperature are rather similar throughout the entire year. However, at 15 m depth, the model does not take into account the heat loss from sediment during ice-cover, which cause an increase in bottom water temperature. During winter, there is a clear mismatch between simulated and observed water temperature. For this reason, all data collected between 1 December - 31 March are excluded from the temperature data used

30 for model calibration and only data between 1 April and 30 November are used for model calibration. Yours is a valuable comment and we better clarified this limitation in our Methods and Discussion.

**Authors' response to Referee 2**

We would like to thank Referee 2 for the comments provided on our manuscript. We added three of the references (Vincent

35 2009; Skowron 2017; Sadro et al., 2019) proposed by the referee in our revised manuscript. Moreover, we improved the X and Y axis description following the referee's suggestions.

**Authors' response to Referee 3**

We would like to thank Referee 3 for the valuable comments and criticisms on the manuscript. The detailed

40 comments provided will be certainly useful to improve the overall quality of this work. Before answering to the specific comments, however, we would like to better clarify our vision of the manuscript. We do not agree with the referee on the fact that our work represents only a case study application. On the contrary, we described an effective methodology that is able to reconstruct historical lake water temperature that can be applied and extended to many

other lakes, not only Lake Erken specifically. Therefore, we believe that this study advances scientific progress. We also think, however, that the specific comments provided by the referee are extremely helpful to better elucidate the general purpose of this work. Please, see below for our responses to the specific comments.

**Referee (1) – Page 2, lines 5-8.** I do not understand the claim that these studies do not use observations to validate their
5  models. This is simply not true. Stefan et al., 1998, include a section called model adequacy tests; Taner et al., 2011, state that they use a previously calibrated model (validation results not shown); and Winslow et al., 2017, include a section called technical validation.

Indeed it would be extremely bad practice to parameterize a model without calibration-validation. Please make your intentions and motivations for the study here much clearer.

10  This would also be improved by expanding the current introduction. At present there are only two paragraphs which cover very little literature. There is a need to root this work within the wider research.

**Authors' response (1) – Page 2, lines 5-8.** This is bad wording on our part. We did not mean to imply that the mentioned studies did not validate their models. What we were trying to say was that these studies have focused on simulating future
15  changes in lake thermal structure (with model validation to present conditions) while in this paper we are advocating for running simulations farther back in time than is normally done for validation purposes in order to provide evidence that climate change has already affected lake thermal structure.

However, the introduction has been rewritten and the references to these papers have been removed. The new first paragraph of the introduction is the following:

20  "*Changes in the thermal structure and mixing regimes of lakes are connected to changes in several climatic factors such as air temperature, solar radiation, cloud cover, wind speed and humidity (Woolway and Merchant, 2019). The alteration of lake hydrodynamic properties has consequences on lake chemistry, biology and ultimately on the ecosystem services that lakes provide (Adrian et al., 2009; Vincent, 2009). Since climatic conditions have changed markedly in the last century and they are expected to change considerably in the next decades (IPCC, 2013), the importance of evaluating how freshwater bodies are*
25  *affected by climate change becomes evident. A direct assessment of how lakes have already been affected by climate change is to analyse historical trends in lake water temperature data. However, the availability of long-term data of lake water temperature is still scarce. For example, there are very few lakes around the world with a long-term record (defined here as >50 years) of water temperature profiles (e.g. Jankowski et al., 2006; Skowron, 2017). Instead, the availability of long-term historical data (>50 years) is often limited to surface water temperature of one or few lakes (e.g. Livingstone and Dokulil,*
30  *2001; Kainz et al., 2017) and the time frame of surface temperature data available for the majority of lakes is limited to 2-3 decades. For example, Sharma et al. (2015) compiled a worldwide database with lake surface water temperature between 1985-2009. The same time frame was used by Schneider and Hook (2010) that reported an average warming trend of 0.045 ± 0.011°C/year of lake surface water temperature in 167 large lakes (>500 km2) using satellite-derived measurements; similarly*

*O'Reilly et al. (2015) reported an average warming trend of 0.34 °C/decade for lake summer surface water temperature in 235 lake worldwide retrieved from both in-situ and satellite data. Even though these studies have demonstrated a rapid warming trend among lakes, the analysis of only surface water temperature is not sufficient to obtain a complete evaluation of the changes in the thermal structure that encompass, for example, temperature trends in the water column and phenology*

5 *of thermal stratification. Moreover, the scarcity of water temperature data before 1980s it difficult to assess earlier thermal conditions for the majority of lakes. A longer record of historical data (> 50 years) provides more background information, allows better documentation of the changes that have already taken place, and leads to more accurate predictions of lake thermal conditions in future decades. One of the best arguments to counter climate change sceptics is well documented long-term records of the ongoing effects of climate change."*

**Referee (2) – Page 4, lines 1-22.** How much data is actually missing from the dataset for each parameter? A table, or similar, detailing the quantity and quality of the data would be extremely helpful.

Why are the additional sites only considered when there is missing data? Are the limited stations used truly representative of conditions across the entire lake? Would the coverage and overall consistency of the observed data not be improved if the

15 same data was used at all times? Please clearly justify your decision-making here - deliberately excluding valid data is problematic.

Similarly, please indicate the locations of the additional stations on your map. It would be useful if the reader could understand the locations of these additional stations relative to the three detailed.

20 **Authors' response (2) – Page 4, lines 1-22.** A detailed description of the number of missing data is available in the supplementary material (tables 1-4). We put these tables in the supplementary material for a better readability of the paper. However, we added a summary table in the manuscript (table 1) with the number of data retrieved from different station for each parameter.

Our meteorological data are either collected from a small island (fig1a, letter B) in the lake or from a meteorological station

25 only a few hundred meters from the lake shore (fig 1a, letter B). Given the station locations, we considered this ideal data for forcing a lake model and it was our assumption that these data should be used when available. When data were missing, we found that the neural network models made use of as many of the surrounding data sources as possible providing the most accurate replacement values. We do not believe that we were excluding valid data, based on our belief (and we suspect a widely held belief) that locally collected data would be most appropriate for modeling. Data from additional sites was only

30 used as a substitute when the most valid data were not available. We added a map showing the location of the additional stations in the manuscript (fig.1)

**Referee (3) – Page 5, lines 13-15; 21-24.** The authors appear to use the same data for calibration-validation. Why is this? Please justify - the standard is to employ a split-sampling approach. Further, the aim is to minimize the variance in the GOF statistics across the calibration-validation period. Without defined periods, you cannot determine the consistency of the model performance.

L21 - What is meant by best? Was the algorithm run multiple times? How is the best one determined when three GOF statistics are used? Please clarify.

You introduce figures 2-4 but provide no further commentary on these. There is no clear discussion with regards to how this indicates good performance. Indeed, you do not refer to your GOF statistics through these figures at all. The reporting of the model performance needs to be significantly expanded. Please also consider reporting model performance per month and/or season - this may help to give insights into whether the model performs worse immediately following the ice-cover period.

Please also note that Figures 3 and 4 do not actually add anything to the reporting of model performance - they give no indication of the GOF of the model. Additionally, the use of inconsistent x-y scales is bad practice and misleading. If producing the figures in R then it is possible to fix the axes across plots/facets.

As a more minor comment - it is not necessary to define the three equations, tehy are standard mathematical equations. What is more important is to explain why these are relevant - what insight does using these GOF statistics provide?

**Authors' response (3) – Page 5, lines 13-15; 21-24.** We agree that for typical applications of models where the goal is to make simulations to future or otherwise different conditions than are covered by the record of measured calibration data it is appropriate to employ a split calibration and validation strategy. However in our case the goal was not to simulate outside of the period of available calibration data, but to use the model to provide a complete and consistent record over a period in which calibration data were available but incomplete (especially in the earlier part of the record). In such a case we believe it is better to make full use of all measured calibration data rather than removing some for a separate validation run. This should ensure that the calibration encompasses the widest possible range of variability and provides parameter values that are most appropriate for the entire period simulated in our study.

L21 - In the ACPy calibration the best set of parameter is calculated by minimizing the log likelihood function. We have now added a specific reference to this in the text of the manuscript.

We reported the model performance in figures 3-4.

The use of inconsistent x-y scales on fig. 3-4 are now. Now all years are shown in the figures.

We removed the standard mathematical equation in the revised version.

**Referee (4) – Page 5, lines 26-28.** Please consider expanding on the limitation of ice-cover - perhaps in the discussion? For example, it would be helpful to suggest how this might be addressed, being unable to account for almost six months of the

year is problematic. Similarly, this should be acknowledged in the section where you describe GOTM. For instance, why do you not simply use a model that does include an ice-cover module given the length of time the lake is ice-covered?

**Authors' response (4) – Page 5, lines 26-28.** The GOTM model used for the simulations documented here did not have a functioning ice model, but instead cut off surface heat exchange when the simulated surface water temperature became negative. This provided a very simple way to make continuous simulations that include freezing conditions that would normally lead to the formation of ice. However, the temperature profiles during winter were not realistic, and could not be used for model calibration. This can be seen in figures 1-2 (below) where a comparison between simulated and observed water temperature at 1 m and 15 m depth is shown for year 2009. At 1 m depth, simulated and observed temperature are rather similar throughout the entire year. However, at 15 m depth, the model does not take into account the heat loss from sediment during ice-cover, which cause an increase in bottom water temperature. During winter, there is a clear mismatch between simulated and observed water temperature. For this reason, all data collected between 1 December - 31 March are excluded from the temperature data used for model calibration and only data between 1 April - 30 November are used for calibrating the model. From the example year shown below (and all other years not shown) it is evident that the measured water temperature quite closely matches the simulated temperature during the period used for calibration. Furthermore, the onset and loss of stratification always falls within this period (1 Apr – 30 Nov), showing that the lack of a fully functioning ice model will not influence simulated estimates of the timing and duration of thermal stratification. Figures 1-2 described here have now been added to the supplementary material. We expanded the description of the limitation of GOTM to simulate ice-cover in section 2.2. Besides that, we analyzed observed Lake Erken ice-cover data between 1941-2017. The results have now been added in the manuscript. The discussion have been now expanded with a paragraph describing ice-cover dynamics at Lake Erken.

Moreover, the reason we used the GOTM model is that this model was also used within the PROGNOS project (http://prognoswater.org/) to provide real-time predictions of water quality using short-term weather forecast data. In this study, the GOTM model which was already set up and tested for Lake Erken as part of PROGNOS, was used here for a different application, namely simulating long-term changes in the lake thermal structure. This model has the advantage that it can be coupled to biogeochemical models, which is crucial for the aims of PROGNOS.

[Figure]

**Figure 1**

[Figure]

**Figure 2**

**Referee (5) – Page 9, figure 5.** Again, please use consistent y-axes and begin at zero. This is bad practice and misleading. Limit the x-axis to the start and end-year.

A continuous line should not be used to represent point data (single seasons per year). This data should be represented as points, or as a single continuous line containing all months.

Finally, please add space between the figures and their titles - at present it looks like the plot titles are related to the dashed line. Including the dashed and solid line in the legend would help. Three duplicated legends are not necessary, replace with a single legend.

**Authors' response (5) – Page 9, figure 5.** Figure 5 ha been improved following your suggestions

**Referee (6) – Page 2, line 9.** Define what is meant by a long record, for hydrological modelling of rivers this would mean > 30 years, for hydroecology > 15 years is considered long.

**Authors' response (6) – Page 2, line 9.** From our point of view the definition of long record is somewhat arbitrary. We think that a long record should encompass the historical changes in climate that have already occurred. For example, a record of 50 years of data can be considered long.

**Referee (7) – Page 1, line 6.** Need to explain what the abbreviation is - for example, consider: "General Ocean Turbulence model (GOTM), a hydrodynamic model configured in Lake Mode".

**Authors' response (7) – Page 1, line 6.** Thank you for the suggestion. This has been added in the revised manuscript

**Referee (8) – Page 2, lines 10-11.** Please provide a citation if making a claim such as this.

**Authors' response (8) – Page 2, lines 10-11.** This is based on the experience of one author (Don Pierson) who has worked for public water utilities and found that documenting the effects of climate change that have already occurred adds support for policies that mitigate future expected changes. To our knowledge, no reference is available.

**Referee (9) – Page 2, line 17.** What does significant mean? How much? Can you give a percentage or some other kind of numerical indication?

**Authors' response (9) – Page 2, line 17.** The word "significant" has been changed to "large". In the revised manuscript, we added the following sentence in section 2.5 (lines 12-13): "*The total number of observed water temperature data in Apr.-Nov. between 1961-2017 was 103454. The number of days with at least one single observed measurements was 6674 days between 1961-2017.*"

**Referee (10) – Pages 2-3, lines 29-30; 1-2.** Extremely limited detail on why the lake was considered. Why should the reader care about the results from this particular work? What is interesting about it?

More information on the case study would also be useful. For example, an overview of the average climate, seasonality, the ecology of the area and anthropogenic influences.

**Authors' response (10) – Pages 2-3, lines 29-30; 1-2.** Lake Erken has been extensively studied in the last 70 years and it has a considerable amount of water temperature data available, which made it a good study case for testing the methodological approach of this paper. Moreover, we think that this paper describes an important methodology to reconstruct complete records of past water temperature of lakes using readily available meteorological data. Thus, the relevance of our work is  not only related to the ecological importance of Lake Erken in itself.

We clarified this point in the introduction of the revised paper. Moreover, we expanded the lake description following your suggestion.

**Referee (11) – Page 3, line 2.** What months represent winter? The reader cannot tell how many months the lake is actually ice-covered. Also, please clarify if it is the entirety of the lake which is ice-covered.

**Authors' response (11) – Page 3, line 2.** In this paper, we considered the period of Dec-Mar as winter,  and it is during this period when the lake is ice-covered. However, in some years, the onset of ice-cover starts in January or later and it also occasionally ends in April. Yes, the lake is ice-covered in its entirety. This information is added in the revised manuscript

**Referee (12) – Page 3, lines 9-12.** Repetitive - could simplify to say: "The model utilises six of these climatic parameters (excluding DP) at an hourly timestep; DP is input on a daily timestep."

**Authors' response (12) – Page 3, lines 9-12**. Thank you for the suggestion. This has been modified following your suggestion.

**Referee (13) – Page 3, lines 4-13.** More information is required for GOTM. Why choose this model specifically? Why is it well-suited for this application? Please also describe the structure of the model, what key processes does it capture? Define and describe the parameters of the model (Table 1). What are the limitations of the model?

5    It is also worth stating that GOTM, and all the other software/codes used, are Open Source.

**Authors' response (13) – Page 3, lines 4-13.** GOTM is mainly used as a stand-alone model for hydrodynamic applications in natural water, and simulates processes such as surface heat fluxes, surface mixed-layer dynamics and stratification processes. The adjusted model parameters in this study are non-dimensional scaling factors affecting the heat-flux, shortwave radiation

10   and wind which are adjusted to minimize the difference between observed and modelled temperature. The minimum turbulent kinetic energy (k_min) and the e-folding depth for visible fraction (g2) are parameters that strongly influence the vertical distribution of light and temperature in the water column. Low values of g2 represent a higher extinction coefficient promoting higher surface temperature.

A known limitation of the model is the lack of an ice-module and a complete energy balance of the ice including ice growth

15   and ice decay is not calculated by GOTM at this time.

We expanded the description of GOTM and its model parameters. We added that GOTM is open source.

**Referee (14) – Page 3, figure 1.** I am aware that the images used for review are not the final high-resolution images. However,

20   this map looks equivalent to a screenshot. A north arrow and, critically, a scale bar, are missing. Additionally, labelling of features such as the roads and the island are unnecessary. Please consider producing a map using GIS Or similar software (mapping options are available in R). A map of Sweden indicating the location of the lake, which would highlight the relative scale, are also necessary.

25   **Authors' response (14) – Page 3, figure 1.** The figure has been substituted with a better quality map following your suggestion.

**Referee (15) – Pages 3-4, lines 15-17; 1-3.** Inconsistent use of meteorological station and weather station - please be consistent. For conciseness, the authors could simply state: "Driving climatic parameters were retrieved from meteorological

30   stations at…".

**Authors' response (15) – Pages 3-4, lines 15-17; 1-3.** The naming is now consistent in the revised manuscript

**Referee (16) – Pages 3-4, lines 15-17; 1-3.** Clearer signposting is required, please refer to the letters that each station represents in the main body text.

**Authors' response (16) – Pages 3-4, lines 15-17; 1-3.** A reference to the point in the map is now in the revised manuscript.

**Referee (17) – Page 3, line 15.** Primarily retrieved from? What does primarily mean specifically?

**Authors' response (17) – Page 3, line 15.** We modified the sentence as follows: "*Driving meteorological parameters were retrieved whenever possible from the Erken laboratory meteorological station…*"

**Referee (18) – Page 1, line 9.** Real is not very clear - consider replacing with "observed" (or similar).

**Authors' response (18) – Page 1, line 9.** Thanks for the suggestion. This has been corrected.

**Referee (19) – Page 3, line 16.** Is the Malma weather station the Erken laboratory meteorological station? This inconsistency is reflected in the caption as well.

**Authors' response (19) – Page 3, line 16.** Yes, it is. This is now clarified in the revised manuscript.

**Referee (20) – Page 4, lines 10; 21.** What is meant by best? Please clarify how this is judged.

**Authors' response (20) – Page 4, lines 10; 21.** We removed lines 8-14 (page 4) and rewrite them as follows: "*To make maximum use of data from surrounding stations we used Artificial Neural Network function fitting analysis (ANN nftool) to predict missing meteorological data at Erken. The analysis was carried out using MATLAB version R2017b (MathWorks Inc. Natick, Massachussets)*". We modified lines 20-21 (page 4) into: "*Offsite and local dataset overlap for at leat 8-10 years to get a reasonable number of data to perform ANN function fitting analysis that describes the input-target relationship.*"

**Referee (21) – Page 4, lines 24-25.** Is the lake always ice-free April-November? Additionally, please replace was with "is" - I presume that the ice-free period has not recently changed, therefore this should be in the present-tense.

**Authors' response (21) – Page 4, lines 24-25.** This period is usually longer than the total period of ice cover which can be variable from year to year. There are occasions when ice continues into April, but the April to November period is definitely representative of ice-free conditions and using data from this period to calibrate the model will definitely avoid errors associated with GOTM's simplistic simulation of ice cover. During the revision of the manuscript we noticed that water temperature data of April and November were not used when calibrating the model. We re-ran the model calibration including April and November measured water temperature and we updated the calibration parameters. The new calibration provided very similar results to the calibration showed in the discussion paper. These are the values of the updated calibrated parameters and model statistics

- Heat-flux factor: 0.863009
- Short- wave radiation factor: 0.970753
- Wind factor: 1.28701
- Minimum turbulent kinetic energy: 1.64873e-06
- e-folding depth for visible fraction: 2.63732

- ln Likelihood: -60469.715
- Bias (°C): -0.04707
- MAE (°C): 0.7529
- RMSE (°C): 1.089
- Correlation: 0.9717

The tables are now updated with these new values in the revised paper.

The sentence is now corrected with the present- tense.

**Referee (22) – Page 4, lines 24-32.** Why is this text part of model calibration? This is still text relating to the input data. Perhaps consider combining 2.3 Data sources of driving parameters-2.3 Model calibration (paragraph 1) into a single Data section.

Additionally, there is an issue with the section numbering: 2.3-2.4-2.3

**Authors' response (22) – Page 4, lines 24-32.** We renamed the section "2.3 Data sources of driving parameters"(page 3, line 14) into "2.3 Data sources of meteorological parameters" and renamed the section "2.3 Model calibration" (page 4, line 23) into "2.5 Observed water temperature data and model calibration".

We corrected the section numbering in the revised manuscript.

**Referee (23) – Page 5, lines 1-2.** As with the hydrodynamic model, the reader needs to know why this approach is used. What is the rationale?

**Authors' response (23) – Page 5, lines 1-2.** We added the following sentence in the text: **"***ACPy is a utility that eliminates the need for time consuming manual calibration of hydrodynamic and water quality models. This allows for more extensive testing and evaluation of model calibrations, ultimately providing more accurate and repeatable results".*

**Referee (24) – Page 5, lines 5-7.** These lines are unclear, please consider rewording.

**Authors' response (24) – Page 5, lines 5-7.** We modified lines 5-7 as follows: "*Simulations were run between 1961 and 2017 but in order to obtain stable initial conditions the model was run over an additional one year spin up using a copy of the 1961 data. In this way, 1961 data were both used as a spin-up year and then reused in the proceeding calibration.*"

**Referee (25) – Page 5, lines 8-13.** The authors state that an algorithm is used in the parameterization. What is the stopping criteria? How does the algorithm select a parameter set?

**Authors' response (25) – Page 5, lines 8-13.** ACPy was set to run 10000 simulation during calibration to get a stable solution to obtain the optimal parameter set that minimizes the log likelihood function. Manual testing found that additional simulations added very little if any improvement to the model calibration.

**Referee (26) – Pages 5-6, lines 28; 1-3.** Please explain to the reader why they should care about these metrics - why are they important? What do they indicate?

**Authors' response (26) – Pages 5-6, lines 28; 1-3.** We rephrased lines 26-27 (page5) as follows: "*We summarized the model temperature output by calculating a number of statistics that can qualify the ecological consequence of changes in thermal stratification using the Lake Analyzer R Package. The ecological implications of the changes of these metrics due to climate change are discussed in detail in the Discussion section".*

**Referee (27) – Page 6, line 12.** Did you test for autocorrelation? Was it all autocorrelated? Please be clearer.

**Authors' response (27) – Page 6, line 12.** Yes, we tested for autocorrelation using acf and pacf function in R. This is now mentioned in the methods (section 2.6). In table 4 are now mentioned which datasets were autocorrelated.

**Referee (28) – Page 6, lines 13-14.** Please correct Figure 3 accordingly - the time-series should not extend beyond the point for which it is useful!

**Authors' response (28) – Page 6, lines 13-14.** This has been modified in the revised manuscript following your suggestion.

**Referee (29) – Page 1, lines 10-12.** Suggest the author's state why the results are split into these sub-intervals; until very late on the paper I presume the split was because pre-1988 records were patchy.

**Authors' response (29) – Page 1, lines 10-12.** Thanks for the suggestion. We added a sentence in the abstract that specifieed that the splitting was selected because of an abrupt change in air temperature.

**Referee (30) – Page 6, line 21.** Please consider moving this line to the start of the section. Please also include the package version for Lake Analyzer (the citation seems relatively old).

**Authors' response  (30) – Page 6, line 21.** The line is now  at the beginning of the section. The package version used here is 1.11.4 and the citation has changed with the following: "Winslow, L., Read, J., Woolway, R., Brentrup, J., Leach, T.,Zwart, J., Albers, S., and Collinge, D: rLakeAnalyzer: Lake Physics Tools. R package version 1.11.4. https://CRAN.R-project.org/package=rLakeAnalyzer, 2018."

**Referee (31) – Page 9, line 2.** What data did this use? The pre-1988 data which included data from mixed stations and the post-1988 data which was much more consistent? Can this finding be trusted?

**Authors' response (31) – Page 9, line 2.** Yes, pre-1988 data mostly included data from mixed stations. However, these data have been adjusted to better represent Lake Erken local conditions (at Malma met station) using Neural Network function fitting analysis. This analysis report a very good agreement between Erken air temperature (target data) and output data (see suplementary material). Given that, we think that we think that air temperature dataset used in this study is reliable.

**Referee (32) – Page 9, line 2.** In the discussion, please explain to the reader why this matters, what it indicates etc. - It is not made clear.

**Authors' response (32) – Page 9, line 2.** From our point of view, an abrupt change in air temperature support the fact that a
5  more rapid change in water temperature is occurring in the last decades and that the effect of climate change on thermal conditions is accelerating. We added the following sentence in the revised paper: *"The Pettitt test showed that a significant abrupt change in annual mean air temperature occurred in 1988 ($p < 0.001$). Therefore, in addition to checking for trends in lake thermal structure over the entire simulation period we also evaluated the possibility of trends occurring over the subinterval 1961-1988 and 1989-2017, and we tested whether a more rapid change in water temperature is occurring after*
10  *1988, following a step-change in annual mean temperature."*

**Referee (33) – Page 9, line 5.** Please define your terms, e.g. epilmnetic.

15  **Authors' response (33) – Page 9, line 5. The terms** epilimnetic and hypolimnetic  are now defined in the revised paper.

**Referee (34) – Page 9, lines 9-14.** Please be consistent in the number of significant figures for temperature.

**Authors' response (34) – Page 9, lines 9-14.** Thanks for the suggestion. This will be corrected in the revised paper.

**Referee (35) – Page 9, line 14.** Please start a new paragraph before discussing thermal stratification.

**Authors' response (35) – Page 9, line 14.** Thanks for the suggestion. This is now corrected in the revised paper.

**Referee (36) – Pages 9-10.** As a decadal mean, it would be useful to see the reporting of confidence intervals for these values. Perhaps consider a table of results.

30  **Authors' response (36) – Pages 9-10.** Thanks for your suggestion. A table of results is now added to the revised manuscript.

**Referee (37) – Page 11, lines 1-8.** You cannot claim that there was a good match. No valid assessment of model performance was provided. This needs to be significantly addressed before such a claim can be asserted.

**Authors' response (37) – Page 11, lines 1-8.** We now referred to the model performance in the revised paper.

**Referee (38) – Page 11, line 12.** I do not agree that it indicates the reliabilty, the wording is too strong. It could be described as a positive indication.

**Authors' response (38) – Page 11, line 12.** We changed the word "reliability" into "consistency".

**Referee (39) – Page 12, lines 1-11.** Much of this text appears to be results.

**Authors' response (39) – Page 12, lines 1-11.** From our point of view, summarizing the major results in the discussion section while at the same time providing a possible explanation for them improves understanding for the reader and also improves the readability of the manuscript.

**Referee (40) – Page 1, lines 10-11; 15-16.** State the months associated with your seasons.

**Authors' response (40) – Page 1, lines 10-11; 15-16.** The months are now added.

**Referee (41) – Page 12, lines 14-17.** The provision of a confidence interval would help to expand upon this further (it could also improve or worsen the difference in results).

**Authors' response (41) – Page 12, lines 14-17.** A confidence interval is not reported by O'Reilly et al. (2015), but only the Sen's slope of lake summer water temperature trends.

**Referee (42) – Page 12, lines 18-22.** Does O'Reilly account for the influence of ice-cover? If yes, could this not also account for some of the discrepancy? Please weight the pros and cons of this study versus theirs accordingly.

**Authors' response (42) – Page 12, lines 18-22.** Trends reported by O'Reilly et al. do not account for ice-cover since the work reports on summer lake surface temperature trends. However, the paper does state that lakes that are always completely icecovered during winter (this is the case of Lake Erken) are experiencing a faster warming trend compared to lakes that do not freeze during winter.

**Referee (43) – Page 13, lines 25-34.** Suggest that the authors consider leading the discussion with this text. At present, it is not clear to the reader why this work or the results are relevant - the implications are not made clear.

**Authors' response (43) – Page 13, lines 25-34**. We consider that the first part of the discussion is useful to demonstrate the validity of the first aim of this work, which is to provide a valuable and reliable method to extend historical water temperature records back in time. In the second part (lines 25-34 page 13 and lines 1-7 page 14) we provided the most important ecological implications that a warmer climate might have on Lake Erken specifically. This order follows the same order of how the aims are described in the introduction.

**Referee (44) – Page 14, lines 8-12.** The assertion of "accurately" cannot be made whilst there is no robust consideration of model performance.

**Authors' response (44) – Page 14, lines 8-12**. We rephrased lines 8-9 (page 14) in the following way: "*The present study has shown that the use of the GOTM model to reconstruct the past 57-years of thermal condition of Lake Erken provided a valuable source of information that could be used to detect changes in its thermal structure. This methodology can be extended to other lakes that have incomplete records of water temperature data. The use of local meteorological data to drive model simulations such as those demonstrated here can be used to extend water temperature records further back in time or fill data gaps where they exist.*"

**Referee (45) – Page 14, lines 13-19.** Suggest that a dedicated conclusion would help to wrap up the paper and reassert the aims/objectives and relevance of the work.

**Authors' response (45) – Page 14, lines 13-19.** A dedicated conclusion is now present in the revised paper.

**Referee (46) – Page 1, lines 23-25.** Abstract does not necessarily make clear why this matters - what is the need for the work?

**Authors' response (46) – Page 1, lines 23-25.** We removed line 5 (page 1) and write as follows: "*Historical lake water temperature records are a valuable source of information to assess the influence of climate change on lake thermal structure. However, in most cases such records span a short period of time and/or are incomplete, providing a less credible assessment of change.*"

**Referee (47) – Page 1, lines 27-29.** This first sentence is repetitive; also not convinced that Samal et al., 2012 is the best citation for this critical statement. There are other more relevant seminal works that the authors may cite.

10 **Authors' response (47) – Page 1, lines 27-29.** We rephrased lines 27-29 as follows: "*Changes in the thermal structure and mixing regimes of lakes are a consequence of changes in several climatic factors such as air temperature, solar radiation, cloud cover, wind speed and humidity (Woolway and Merchant, 2019).*"

15 **Referee (48) – Page 2, lines 1-2.** Again, repetition - it is self-evident that a rise in lake water temperature increases water temperature - please be more concise.

**Authors' response (48) – Page 2, lines 1-2.** Thanks for your suggestion. The citation of Arhonditsis et al. has been removed

**Referee (49) – Page 2, lines 5-7.** It would be helpful to explain what some of the conclusions of these studies are/were - it makes it clearer to the reader why there is a need for this.

**Authors' response (49) – Page 2, lines 5-7.** We now removed the reference to this paper and change the first paragraph of
25 the introduction (see Author,s response 1).

**Referee (50) – Page 6, lines 4-7.** I would like to highlight that the level of description here is excellent and represents the level that should be achieved throughout the manuscript.
30
**Authors' response (50) – Page 6, lines 4-7.** Thanks for your comment!

[revised manuscript text omitted]